# Quantitative analysis of mammalian translation initiation sites by FACS-seq

William L Noderer[1], Ross J Flockhart[2], Aparna Bhaduri[2,3], Alexander J Diaz de Arce[1], Jiajing Zhang[2], Paul A Khavari[2,4] & Clifford L Wang[1,*]

## Abstract

An approach combining fluorescence-activated cell sorting and high-throughput DNA sequencing (FACS-seq) was employed to determine the efficiency of start codon recognition for all possible translation initiation sites (TIS) utilizing AUG start codons. Using FACS-seq, we measured translation from a genetic reporter library representing all 65,536 possible TIS sequences spanning the −6 to +5 positions. We found that the motif RYMRMVAUGGC enhanced start codon recognition and translation efficiency. However, dinucleotide interactions, which cannot be conveyed by a single motif, were also important for modeling TIS efficiency. Our dataset combined with modeling allowed us to predict genome-wide translation initiation efficiency for all mRNA transcripts. Additionally, we screened somatic TIS mutations associated with tumorigenesis to identify candidate driver mutations consistent with known tumor expression patterns. Finally, we implemented a quantitative leaky scanning model to predict alternative initiation sites that produce truncated protein isoforms and compared predictions with ribosome footprint profiling data. The comprehensive analysis of the TIS sequence space enables quantitative predictions of translation initiation based on genome sequence.

**Keywords** FACS-seq; Kozak motif; proteome modeling; start codon; translation initiation
**Subject Categories** Chromatin, Epigenetics, Genomics & Functional Genomics; RNA Biology
**Mol Syst Biol. (2014) 10: 748**

## Introduction

Cells can control protein translation levels by tuning translation initiation (Kozak, 1991; Sonenberg & Hinnebusch, 2009; Ivanov *et al*, 2010). In eukaryotes, translation initiation typically follows the scanning ribosome model. In this model, the ribosomal preinitiation complex consisting of the small 40S ribosomal subunit, Met-tRNA, eIF2-GTP, eIF1, eIF1A, eIF3, and eIF5 is loaded onto the mRNA 5′ cap (Kozak, 2002b; Jackson *et al*, 2010; Hinnebusch, 2011). The preinitiation complex, along with additional initiation factors, scans from the mRNA 5′ cap in the 3′ direction in search of a start codon, which in most circumstances is AUG. When the preinitiation complex recognizes a start codon, initiation factors dissociate and a phosphate group is irreversibly released. The large 60S ribosomal subunit is then able to join the small 40S ribosomal subunit to complete the translation initiation process (Pestova & Kolupaeva, 2002; Nanda *et al*, 2013). Yet, the scanning ribosomal preinitiation complex does not initiate at every start codon that it encounters. With a certain probability, the ribosomal complex initiates translation, while others 'leak' past the start codon and continue scanning. This probability of initiation, or translational initiation efficiency, is governed by the sequence of the translation initiation site (TIS), which consists of the start codon and its adjacent bases. Therefore, cells can control translation levels in a gene sequence-dependent manner by controlling the efficiency at which a ribosome recognizes the start codon and initiates translation.

Kozak has reported GCCRCCAUGG (purine, R=A or G; start codon underlined) to be a highly efficient mammalian TIS (Kozak, 1986, 1987a,b). By further generating point mutants of TIS sequences and evaluating reporter expression, Kozak found the −3R and +4G to be the first and second most important bases for efficient initiation, respectively (+1 denotes first base of start codon) (Kozak, 1986, 1997). Consequently, as a 'rule of thumb', highly efficient TISs are typically considered those with an AUG start codon, a −3R, and a +4G (Kozak, 1995; Harte *et al*, 2012). Applying these criteria, 40% of human genes (8,629) utilize highly efficient TISs (RefSeq release 55, September 2012). Yet, thousands of genes have apparently evolved TISs that are not highly efficient and could have been purposefully tuned for low or moderate translation efficiency (Kozak, 1991; Lukaszewicz *et al*, 2000). In fact, 9.6% of human genes (2,065) utilize neither a −3R nor a +4G. Any quantitative, genome-wide analysis of protein expression would need to account for the broad range of translation initiation efficiencies achieved by

1   Department of Chemical Engineering, Stanford University, Stanford, CA, USA
2   The Program in Epithelial Biology, Stanford University School of Medicine, Stanford, CA, USA
3   The Program in Cancer Biology, Stanford University School of Medicine, Stanford, CA, USA
4   Veterans Affairs Palo Alto Healthcare System, Palo Alto, CA, USA
    *Corresponding author. Tel: +1 650 736 1807; Fax: +1 650 725 7294; E-mail: cliff.wang@stanford.edu

the diverse TIS sequences. To our knowledge, no study has systematically analyzed all possible mammalian TIS sequences.

In summary, we are limited by our knowledge of the relationship between TIS sequence and translation initiation efficiency. Here, we combined fluorescence-activated cell sorting (FACS) with high-throughput DNA sequencing to analyze the translation initiation efficiency of 65,536 TIS sequences. After gauging the translation level mediated by each TIS sequence utilizing an AUG start codon, we report a comprehensive analysis of the TIS motif. We also developed algorithms that accounted for variable TIS-dependent initiation levels throughout the genome. Our analysis revealed key roles for TIS-dependent expression control in regulating cellular processes, generating protein isoforms, and tumorigenesis.

## Results

### High-throughput analysis TIS sequences by FACS-seq

Our first objective was to measure translation levels associated with each possible TIS sequence utilizing AUG as a start codon. Because we sought a high-throughput solution, it was crucial that these measurements could be made easily and precisely. To achieve this objective, we adapted a genetic reporter system that we developed previously (Ferreira *et al*, 2013). With this system, different TIS sequences of interest were used to initiate translation of green fluorescent protein (GFP) (Fig 1A). As a reference, red fluorescent protein (RFP) was translated from the same transcript using an internal ribosome entry site (IRES) and the TIS sequence GCCAC-CAUGGU. Because both fluorescent proteins were translated from the same transcript, normalizing GFP with RFP reduced the effect of extrinsic noise and improved our measurement of translation (Supplementary Fig S1) (Dean & Grayhack, 2012). Next, we generated a TIS reporter library. By utilizing a PCR-amplification approach using degenerate primers, we inserted TIS sequences with randomly chosen bases flanking the AUG start codon at positions −6 to −1, +4, and +5 (i.e., NNNNNNAUGNN where Ns indicate varied positions). Our library was sufficiently large so that it likely (> 99% probability) contained all 65,536 ($4^8$) TIS sequences.

In the past, we have used flow cytometry to analyze individual TIS sequences one genetic construct at a time (Ferreira *et al*, 2013).

However, because of the sheer number of possible TIS sequences, we could not take this approach. Instead, we stably transduced cells (PD-31 mouse pre-B lymphocytes) with the entire TIS reporter library. This produced a heterogeneous population where each cell received one copy of the reporter construct. We now sought to generate a population histogram that reflected the translation level for each TIS sequence in the library. Using FACS, we isolated cell subpopulations producing GFP/RFP levels within 20 different ranges (a.k.a. FACS gates), which correspond to the *x*-axis intervals of the desired histograms (Fig 1B). To reduce sorting time while maintaining adequate analytical resolution, the 20 gates were chosen such that 5% of the total population fell into each gate. After sorting,

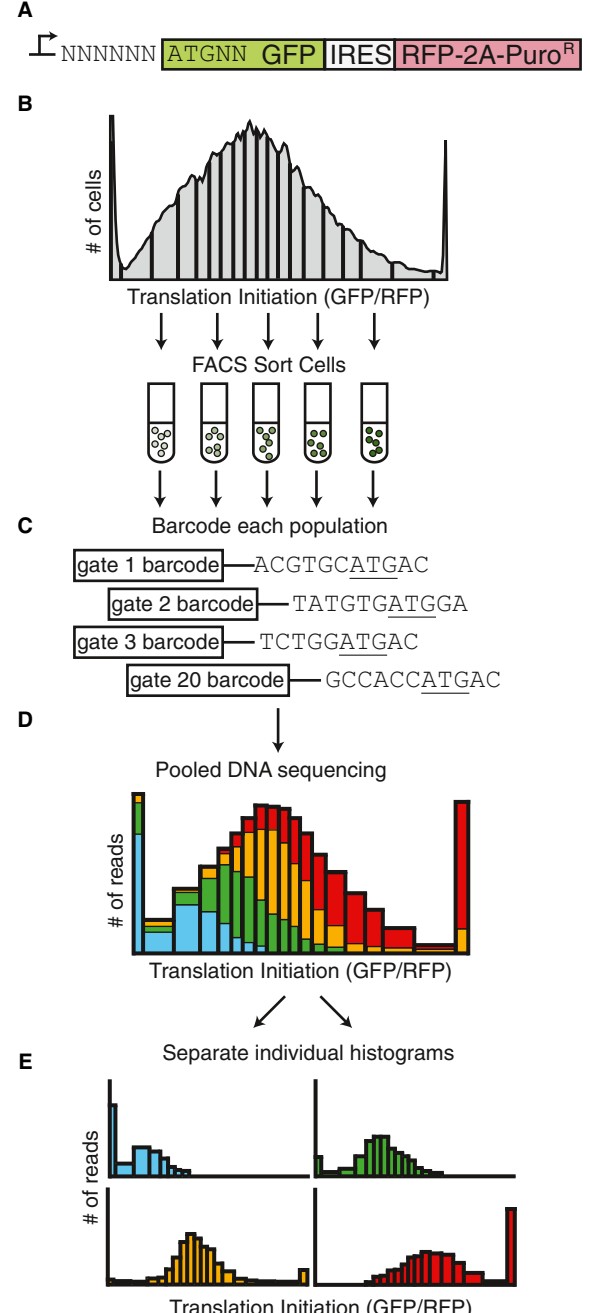

### Figure 1.   Analysis of TIS sequences via FACS-seq.

A   A TIS reporter library was created by degenerate PCR to create a random permutation of the six bases upstream and two bases downstream of a GFP start codon, resulting in a library size of 65,536 TIS sequences. An IRES followed by RFP was used to normalize the GFP expression.

B   Stably transduced cells were sorted based on translation efficiency Analysis of TIS sequences via FACS-seq (GFP/RFP). 20 gates were drawn such that each gate contained 5% of the total library population. Cells that were off-scale were collected in the last gate.

C   The TIS sequences from each sorted population were PCR amplified and barcoded.

D   The barcoded TIS library was pooled and sequenced. The number of reads for each TIS sequence and barcode combination was counted.

E   An individual histogram for each TIS sequence is created *in silico*.

Data information: green fluorescent protein, GFP; red fluorescent protein, RFP; internal ribosomal entry site, IRES; 2A slippage site, 2A; puromycin resistance gene, Puro[R].

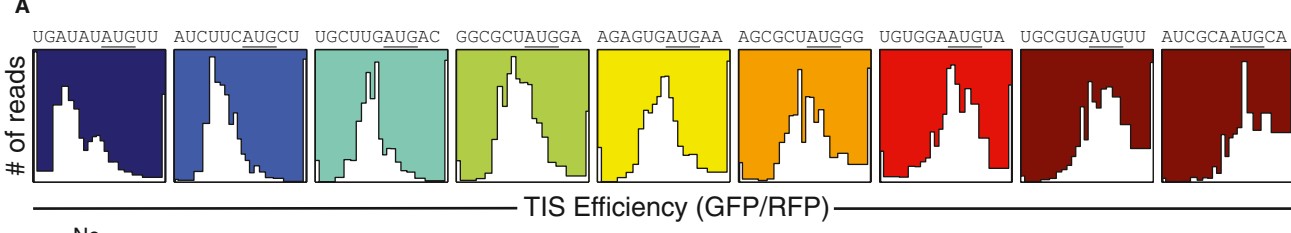

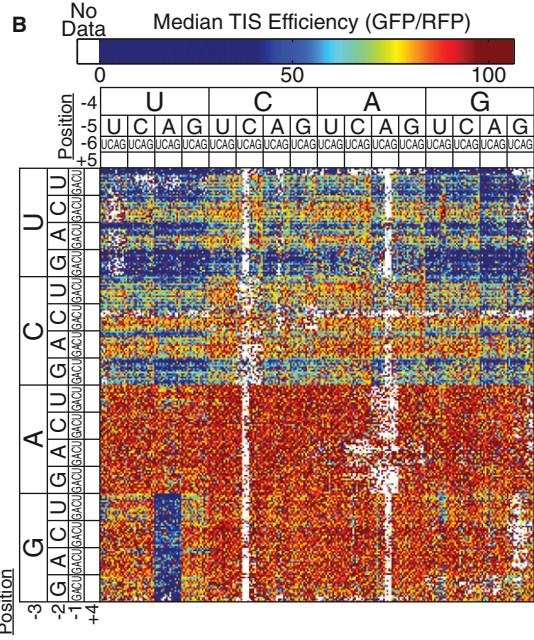

**Figure 2.  TIS efficiency determined by FACS-seq.**

A  Individual FACS-seq histograms spanning a range of translation efficiency values. The background color corresponds to the median translation initiation efficiency. The TIS sequences are indicated above each histogram.

B  Heat map of entire TIS sequence space. The sequences were arranged to highlight trends in TIS strength. The +4/+5 positions were not labeled, but they follow the same nucleotide order of U/C/A/G.

each subpopulation still contained a mixture of cells with different TIS reporters, but now the numbers of each specific TIS sequence in each subpopulation could be used to generate population histograms for each translation level. To determine the numbers of each TIS sequence, we PCR amplified the vector region containing the TIS sequence from each subpopulation (Fig 1C). Using PCR, we also added a barcode sequence that could be used to identify subpopulations. All amplified products were then pooled and sequenced on the Illumina next-generation sequencing platform (Fig 1D). We counted the number of each TIS sequence for each translation interval and generated histograms describing the translation level associated with each TIS sequence (Fig 1E). We refer to this combined FACS and high-throughput sequencing approach as FACS-seq.

From each of these population distributions (i.e., histograms, Fig 2A), we determined a median translation level (GFP/RFP). These values were then normalized so that translation of 100 relative units was equal to that produced by the TIS GCCACCAUGGG, a TIS often cited as being optimal for initiation (Kozak, 1987a; Harte *et al*, 2012). Since the TIS reporters only differed at the TIS sequence, we used reporter output as a measure of the relative translation initiation efficiency of each TIS sequence; however, when interpreting our results, one should still consider that changes to the eight TIS bases could have indirectly affected the many other factors that influence translation, for example, loading of the ribosome preinitiation complex or the rate of translation elongation. To visualize all of the relative translation initiation efficiencies measured by FACS-seq, we generated a heat map (Fig 2B) where

sequences were clustered to highlight trends in TIS efficiency. Immediately, two trends supported the validity of our FACS-seq method. The strongest trend could be seen when comparing the translation efficiencies of TIS with a purine (A or G) to those with a pyrimidine (C or U) at the −3 position (Fig 2B; bottom vs. top halves, respectively). This confirmed previous reports that −3 purines promote efficient translation initiation (Kozak, 1986, 1987a, 1995). Second, when the GFP TIS sequences contained an upstream, out-of-frame AUG start codon, it would be expected that ribosomes preferentially initiate translation at the upstream AUG and synthesize an out-of-frame peptide instead of GFP (Kozak, 1995; Calvo *et al*, 2009). In agreement with this prediction, the GFP TIS sequences with −5A/−4U/−3G demonstrated significantly reduced translation, distinguished by the blue block of values near the bottom-left corner of the heat map (Fig 2B).

## Modeling the relationship between TIS sequence and initiation efficiency

Ideally, the sequence coverage generated by FACS-seq would be sufficient to generate histograms of high resolution for all TIS sequences. In practice, TIS sequences with repeat regions were sometimes absent or underrepresented (Fig 2B, white values). Because of data noise, we also could not be certain that any single value could precisely represent an initiation efficiency. Analogous to data obtained from microarray analysis of mRNA, the raw data revealed meaningful trends but individual data points may or may

not stand on their own (Kerr *et al*, 2000; Sultan *et al*, 2002). By fitting the raw data to a model, our goal was to more accurately estimate the TIS efficiency for every TIS sequence.

We first attempted to model the relationship between TIS sequence and efficiency with a mononucleotide position weight matrix (PWM), a model where each base contributes independently to initiation without cooperativity. The mononucleotide PWM was constructed by performing regression analysis on the natural logarithm of the raw FACS-seq data (Fig 3A and B) (Barrick *et al*, 1994; Salis *et al*, 2009). To avoid multiple initiation sites, TIS sequences containing an upstream AUG (e.g., TISs containing −5A, −4U, and −3G) were excluded from our training dataset. The resulting mononucleotide PWM revealed many of the known qualitative trends in TIS efficiency. For example, the mononucleotide PWM predicted that a −3A enhanced TIS efficiency by 58% relative to a −3U (Kozak, 1986, 1987a). To test the mononucleotide PWM, we compared the TIS efficiencies predicted by the model with those determined by conventional flow cytometry of cells expressing individual TIS reporters (i.e., each cell population translated GFP using a specific TIS sequence). We found that the mononucleotide PWM only moderately improved the accuracy of the TIS efficiency predictions relative to the raw FACS-seq data (raw data, $R^2 = 0.44$, $P = 2.0 \times 10^{-6}$; mononucleotide PWM, $R^2 = 0.52$, $P = 1.5 \times 10^{-15}$) (Fig 4A and B).

Because the mononucleotide PWM accounts for base contributions independently and does not consider base combinations, this treatment might be an inadequate model of the ribosome–mRNA interactions (Bulyk *et al*, 2002). To investigate the possibility of cooperativity between TIS positions, we next evaluated a dinucleotide PWM, which accounts for all possible interactions between any two base positions. The dinucleotide PWM was trained on the raw

FACS-seq data in a similar manner as the mononucleotide PWM, again excluding TIS sequences with an upstream AUG (Supplementary Table S1). Compared to the mononucleotide PWM, the dinucleotide PWM was in better agreement with the test data generated by conventional flow cytometry ($R^2 = 0.83$, $P < 2 \times 10^{-16}$) (Fig 4C). The model also more accurately predicted TIS efficiency values for those TIS sequences absent from the raw FACS-seq data (Fig 4C, filled squares). The substantial improvement of the dinucleotide PWM over the mononucleotide PWM demonstrated the importance of cooperativity between pairs of bases in start codon recognition. A trinucleotide PWM did not further improve the model ($R^2 = 0.71$, $P < 2 \times 10^{-16}$) (Fig 4D). The poor performance of the trinucleotide PWM may indicate over-fitting of the raw FACS-seq data or may indicate that higher order cooperativity beyond pairwise interactions is of lesser importance. The dinucleotide PWM was used to generate a complete TIS efficiency reference table for every TIS sequence (Supplementary Table S2).

The dinucleotide PWM indicated the strongest interaction to be between the +4 and +5 positions. To visualize this interaction, we grouped TIS sequences according to their +4 and +5 bases and calculated the median translation initiation efficiency for each set of TISs (Fig 5A). In line with the dinucleotide PWM, we observed a substantial interaction between the +4 and +5 positions. For example, TIS sequences containing a +4G/+5C were on average 24.8 ± 0.2% more efficient than sequences containing a +4G/+5A, suggesting that a +5C improved start codon recognition (relative to a +5A). However, TIS sequences containing a +4C/+5C were on average 14.6 ± 0.3% less efficient than sequences containing a +4C/+5A, suggesting the opposite trend. Moreover, a +4G, which is commonly thought to be critical for efficient start codon recognition (Kozak, 1986, 1987a), was not always optimal. TIS sequences containing a +4G/+5U were on average 8.2 ± 0.3% less efficient than sequences containing a +4U/+5U. Due to the strong interaction between the +4 and +5 positions, no individual base at either position was a singular determinant for translation initiation efficiency.

Surprisingly, the dinucleotide PWM predictions depended heavily on the −4 and −2 positions, which were previously reported to have a minimal impact on start codon recognition (Kozak, 1986, 1987a). Moreover, the model revealed a strong interaction between both positions and the critical −3 position. To visualize the effect, we grouped TIS sequences according to their −4, −3, and −2 sequence (Fig 5B and C). In agreement with the dinucleotide PWM, we observed that the influence of the −4 and −2 positions was dependent on the −3 position. For example, TIS sequences containing a −4C/−3A were on average only 1.7 ± 0.3% more efficient than sequences containing a −4G/−3A, suggesting that the −4 position had minimal influence. However, sequences containing a −4C/−3U were 15.3 ± 0.4% more efficient than sequences containing a −4G/−3U, demonstrating that the influence of the −4 position was dependent on the −3 position. A similar cooperativity was observed between the −3 and −2 positions. For example, TIS sequences containing a −3G/−2A were 2.8 ± 0.3% more efficient than sequences containing a −3G/−2G. However, sequences containing a −3C/−2A were 17.6 ± 0.4% more efficient than sequences containing a −3C/−2G. The implication of these results is that an A or C in the −4 and/or −2 positions partially compensates for a U or C in the −3 position (which generally disfavors start codon recognition).

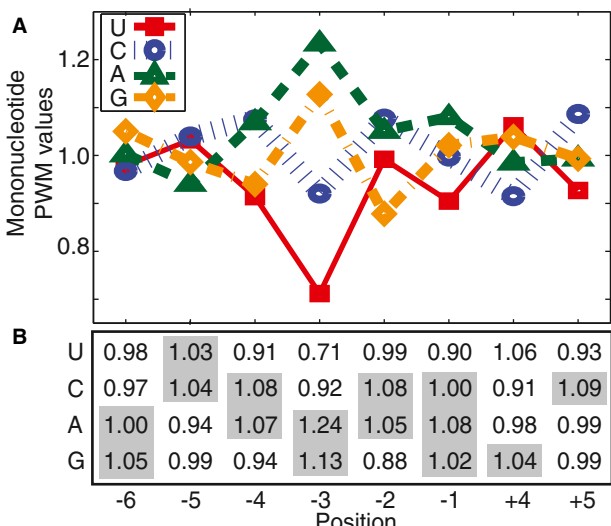

**Figure 3.  Mononucleotide PWM of TIS efficiency.**

A   Normalized values of the mononucleotide PWM ($\exp(C_{b,i})$). Values above 1.00 enhance TIS efficiency; values below 1.00 reduce TIS efficiency.

B   The same values in matrix form. The bases from the high-efficiency TIS motif are highlighted. TIS efficiency is calculated by multiplying the intercept value 76.9 by the appropriate base/position values.

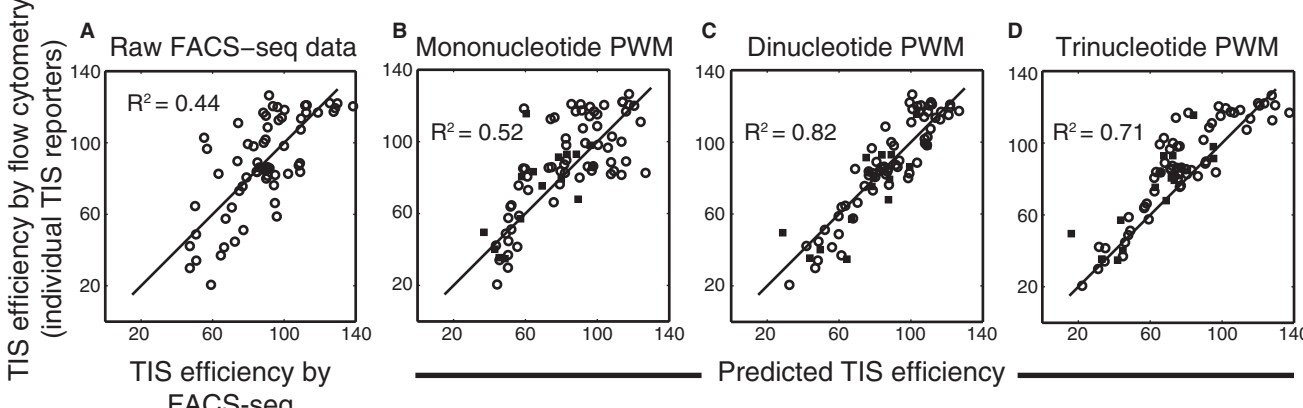

**Figure 4.  Comparison on TIS efficiency models.**

A–D   Individual reporter constructs with TIS sequences of interest were assayed for TIS efficiency (*y*-axis) and compared with (A) the raw FACS-seq data generated from analysis of the degenerate TIS reporter library, (B) the mononucleotide PWM predictions, (C) the dinucleotide PWM predictions, or (D) the trinucleotide PWM predictions. Filled squares correspond to TIS sequences absent from the raw FAC-seq data.

Source data are available online for this figure.

The dinucleotide PWM provided the most accurate, quantitative method of determining TIS efficiency. Yet, for simplicity, we also sought to generalize our results with the following high-efficiency TIS motif:

RYMRMV<u>AUG</u>GC

where Y = U or C, M = A or C, R = A or G, and V = A, C, or G. The −4, −3, −2, +4, and +5 positions were the most critical for efficient start codon recognition. Due to the considerable interactions between the +4 and +5 positions, it was difficult to generalize these positions without accounting for dinucleotide combinations. Nonetheless, we chose to include the +4G/+5C combination because it resulted in consistently high initiation efficiency, even without the −3R. Other +4/+5 combinations, such as +4G/+5G or +4A/+5C, were also favorable to initiation. TIS sequences that matched our full motif were on average 10 ± 1% more efficient than TIS sequences that only contained a −3R/+4G (Supplementary Fig S4).

As opposed to previously reported TIS consensus motifs, which were based on the frequency of TIS usage in the genome, we have constructed a high-efficiency TIS motif by directly measuring the relative translation efficiency of tens of thousands of TIS sequences.

**Effects due to different reporter genes, cell types, environment, and mRNA secondary structure**

We next sought to determine the degree to which other experimental variables affected our measurements of translation initiation. When we evaluated a subset of TIS sequences using our reporter system, we observed reproducible levels of translation initiation across a variety of cell lines: mouse pre-B lymphocytes (PD-31), mouse fibroblasts (NIH-3T3), human colon cancer (HCT-116), human cervical cancer (HeLa), human hepatocellular carcinoma (HepG2), and human chronic myelogenous leukemia (K562) (Supplementary Fig S2A). We also found that translation of yellow fluorescent protein (YFP) and blue fluorescent protein (BFP) yielded

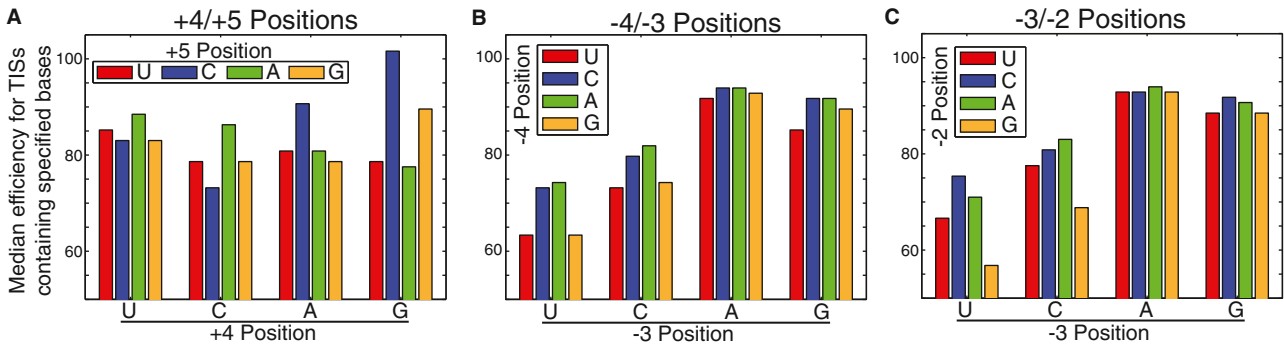

**Figure 5.  Cooperativity in TIS efficiency.**

A–C   Median TIS efficiency (GFP/RFP) for all TIS sequences that contain the specified (A) +4/+5 nucleotides, (B) −3/−2 nucleotides, or (C) −4/−3 nucleotides. All median TIS efficiency values are ± 0.2 (95% CI).

Source data are available online for this figure.

similar but not identical measurements when compared to the translation levels of the GFP reporter (Supplementary Fig S2B). Furthermore, the TIS efficiency measurements were not greatly affected by varying the culture and growth conditions (Supplementary Fig S2C). In summary, while factors other than the TIS sequence can affect translation initiation efficiency, our experiments indicated that our reporter system was adequate to gauge relative levels of translation initiation efficiency across various experimental situations.

We considered whether mRNA secondary structures could have affected the output from our TIS reporters. Secondary structures, especially near the 5′ cap, have been reported to decrease the overall rate of translation, presumably by decreasing the flux of scanning ribosomes (Babendure *et al*, 2006). The 5′ UTR used in this study was predicted to have a weak hairpin at the 5′ cap (bases 1–28, $\Delta G = -8$ kcal/mol) and a stable structure in the retroviral U5 region (bases 83–191, $\Delta G = -51$ kcal/mol). These mRNA structures may have affected the absolute rate of translation. However, because the structures were shared by the entire TIS reporter library, the relative TIS efficiency values were not affected. It was also possible that specific TIS sequences could have resulted in the formation of an mRNA structure near the start codon (Gu *et al*, 2010; Tuller *et al*, 2010; Goodman *et al*, 2013). To investigate this possibility, we calculated the folding energy for 65,536 mRNA sequence associated with each TIS sequence in our reporter library. Each sequence consisted of the 70 bases upstream of the TIS, a single 11-base TIS region, and 70 bases downstream of the TIS (Supplementary Fig S3). The difference between the most stable and least stable secondary structure was −16 kcal/mol. We did not observe any significant relationship between the mRNA folding energy and the TIS efficiency ($P = 0.18$).

**Genome-wide analysis of TIS efficiency**

Next, we performed a quantitative genome-wide analysis of human TIS efficiency. As a point of reference, we first considered the distribution of initiation efficiencies for the entire TIS sequence space (i.e., all 65,536 TIS sequences) as determined by our dinucleotide PWM (Fig 6A). The range of TIS efficiencies varied 12-fold with the distribution skewed toward efficient initiation, indicating that the majority of possible TIS sequences that contain an AUG start codon result in efficient initiation. The TIS sequences predicted to be the most efficient often contained a −3R or a +4G, consistent with previous research (Kozak, 1986, 1987a). However, there were many discrepancies. For example, 15% of the TIS sequences in the top quartile did not have a −3R and 68% did not have a +4G. These discrepancies emphasize the importance of considering the full sequence when determining TIS efficiency.

Next, we analyzed the initiation efficiency for all human protein coding TISs (i.e., TISs that start a protein coding ORF). The dinucleotide PWM was applied to 21,579 protein coding TIS sequences from the RefSeq database (RefSeq release 55, September 2012). By comparing the distribution of the protein coding TISs to that of the entire TIS sequence space, we concluded that human protein coding TISs are enriched for efficient initiation sequences (Fig 6B, Supplementary Fig S5), which qualitatively agrees with previous reports (Kozak, 1986). Quantitatively, the mean efficiency of human protein coding TISs was $15.2 \pm 0.4\%$ higher than that of the TIS sequence space (i.e., higher than if the TIS sequences were

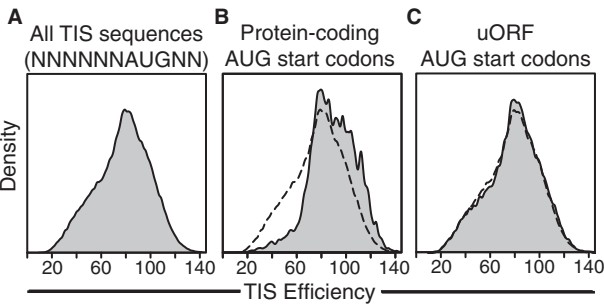

**Figure 6. Genome-wide analysis of human translation initiation efficiency.**

A Distribution of the initiation efficiencies for the entire TIS sequence space (i.e., all permutations of the motif NNNNNNAUGNN).

B Distribution of initiation efficiencies for all annotated human TISs.

C Distribution of TIS efficiencies for all human uORF TISs (i.e., AUG start codons located in 5′ UTRs). Distribution of efficiencies for all TIS sequences included for comparison (dashed-line, B and C). Efficiency values were predicted with the dinucleotide PWM. A value of 100 corresponds to the TIS efficiency of GCCACCAUGGG.

chosen at random). 42% of the protein coding TISs had an efficiency value in the top quartile of the TIS sequence space, while only 7% of the protein coding TISs had an efficiency value in the bottom quartile of the TIS sequence space. We also compared the protein coding TISs to upstream ORF (uORF) TISs (i.e., AUG-containing TISs found in the 5′ UTR that initiate uORFs). The dinucleotide PWM was used to predict the efficiency of 37,474 uORF TIS sequences found in the RefSeq database. The distribution of the uORF TISs was similar to that of the entire TIS sequence space (Fig 6C, Supplementary Fig S5). The mean efficiency of uORF TISs was only $1.6 \pm 0.4\%$ higher than that of the TIS sequence space, suggesting that there was no overarching preference for higher or lower initiation efficiency at uORFs. While protein coding TISs were enriched for efficient initiation and minimal leaky scanning, uORFs used the full spectrum of possible TIS efficiencies. To our knowledge, this is the first quantitative genome-wide survey of TIS efficiency.

**Somatic TIS mutations associated with tumorigenesis**

Our comprehensive analysis of the TIS sequence space enabled us to search for candidate driver mutations in tumorigenesis. Mutations in the TIS motif have the potential to alter the efficiency of start codon recognition, thereby disrupting the protein translation levels and possibly impacting human health (Kozak, 2002a; Wolf *et al*, 2011). Prior to this work, the qualitative effect of TIS mutations at positions other than the −3 or +4 was difficult to predict (Xu *et al*, 2010). However, a few studies had already demonstrated that mutations at positions other than the −3 or +4 are biologically relevant (Afshar-Kharghan *et al*, 1999; Usuki & Maruyama, 2000; González-Conejero *et al*, 2002; Jacobson *et al*, 2005). For example, a −1C > U polymorphism in the *CD40* gene has been associated with Graves' disease, an autoimmune disorder (Jacobson *et al*, 2005). The polymorphism decreased the *CD40* TIS efficiency by 15–32%, demonstrating that even modest changes in TIS efficiency can be biologically relevant. TIS mutations at positions other than the −3 and +4 are largely uncharacterized despite their potential impact on human health.

We screened the Catalogue of Somatic Mutations in Cancer (COSMIC) for somatic TIS mutations in positions other than the −3 or +4. The database contains somatic mutations associated with genes implicated in tumorigenesis (reported in the literature and from sequencing of patient samples) (Forbes *et al*, 2011). We were most interested in the TIS mutations that could be causative drivers of tumorigenesis (in contrast to passenger mutations). Therefore, we chose mutations where the predicted change in protein translation efficiency was consistent with other known tumor expression patterns (i.e., changes in the mRNA level or gene copy number). For example, if a TIS mutation in the database was predicted to decrease the protein translation efficiency and the gene containing that mutation was also known to be down-regulated at the mRNA level in other tumor samples, then the TIS mutation was consistent with known tumor expression patterns.

The COSMIC database contained 554 mutations in the TIS motif (−6 to +5 positions, excluding AUG start codon) (Supplementary Table S3). Of these mutations, 276 were located at positions other than −3 or +4. We assessed the candidate mutations using publicly available gene expression data, gene copy number data, and finally, through a literature search. Here, we have chosen to report 7 TIS mutations that spanned an array of predicted changes to TIS efficiency, which we used to verify our predictions (Table 1). The identified mutations occurred at a variety of positions and were potentially causative in tumorigenesis. The 11-base TIS sequences, both with and without the somatic mutation, were generated, inserted into our dual fluorescence TIS reporter, and analyzed by conventional flow cytometry. The predicted changes in TIS efficiency were in good agreement ($R^2 = 0.93$) with the measured changes in TIS efficiency (Fig 7), indicating that the dinucleotide PWM can be used to predict the effect of mutations in the TIS sequence.

For example, we identified a mutation in the −4 position of the *NOS1* TIS sequence that could play a role in tumorigenesis. The gene *NOS1* is part of the neuronal nitrous oxide synthesis pathway and has been associated with tumorigenesis (Fukumura *et al*, 2006). In lower grade glioma datasets, *NOS1* mRNA expression was on average down-regulated twofold (The Cancer Genome Atlas, https://tcga-data.nci.nih.gov/tcga/). A mutation that decreases the translation efficiency would impact on *NOS1* protein levels in a manner similar to the observed mRNA down-regulation. A −4C > U mutation in the *NOS1* TIS was identified in the COSMIC database and was predicted by the dinucleotide PWM to reduce the translational efficiency by $27 \pm 9\%$. When we analyzed the mutation with specifically generated TIS reporters (with and without the mutation), we observed a $30 \pm 3\%$ decrease in efficiency. As another example, we identified a possible driver mutation in the +5 position of the *MED8* gene, which assists in the regulation of cell proliferation through the transcriptional activation of RNA polymerase II-dependent genes (Miklos *et al*, 2008; Taatjes, 2010). A threefold up-regulation of *MED8* mRNA was observed in uterine tumors and endometrioid samples (The Cancer Genome Atlas). A +5U > G mutation in the *MED8* TIS was predicted to increase the efficiency by $46 \pm 15\%$ and was observed to increase the efficiency of our TIS reporter by $73 \pm 13\%$. Therefore, the mutation was consistent with the tumor expression pattern and was identified as a candidate tumorigenic driver mutation. In summary, we have linked known tumorigenic expression patterns, previously validated only for RNA level or gene copy number, with changes in TIS efficiency, thereby providing a candidate explanation for how the TIS mutations could promote tumorigenesis.

### Employing a quantitative leaky scanning model to predicting truncated protein isoforms

Next, we investigated how suboptimal TIS efficiency could increase proteomic diversity. Ribosomes that 'leak' past a TIS are capable of initiating at downstream TISs, thereby generating alternative translational isoforms. Just as alternative RNA splicing enables a single gene to encode multiple transcript variants, leaky scanning enables a single gene to encode multiple protein isoforms (Ingolia *et al*, 2011; Lee *et al*, 2012; Michel *et al*, 2012; Ben-Yehezkel *et al*, 2013). In the instances where a TIS is downstream and in-frame with the transcript's annotated start codon, an N-terminal truncated protein is translated. Compared to the full-length protein, the N-terminal

**Table 1.**  **Somatic TIS mutations associated with tumorigenesis**

| Gene | TIS sequence and mutation | Tumor expression patterns |
|---|---|---|
| MAP2K3 | UCCUGCAUGUC (A at −6) | Downregulated in human breast cancer. Forced expression induces senescence (Jia *et al*, 2010) |
| MBP | GAUGUGAUGGC (A at +5) | Deleted in 61% of ovarian serous carcinomas and 54% of head neck squamous cell carcinomas ($P < 3 \times 10^{-13}$) |
| NOS1 | AUCUGCAUGGG (U at −4) | Downregulated 2-fold in glioma samples ($P = 3.6 \times 10^{-39}$) |
| PLD5 | ACGUACAUGAA (A at −6) | Downregulated 2.3-fold in serous carcinoma ($P < 3 \times 10^{-13}$) |
| DAB2 | CUUGCCAUGUC (A at +5) | Amplified in 71% of lung squamous cell carcinomas, and 56% of lung adenocarcinomas ($P = 3.2 \times 10^{-4}$) |
| MED8 | GAUCUCAUGGU (G at +5) | Upregulated 3-fold in uterine tumor and endometrioid samples ($P = 4 \times 10^{-101}$). Key regulator of cell proliferation (Miklos *et al*, 2008) |
| DHX33 | GAUGGCAUGCU (U at −5) | Upregulated 2-fold in lung cancer ($P < 10^{-100}$). Ras[G12V] upregulates gene; p19[ARF] downregulates gene (Zhang *et al*, 2013) |

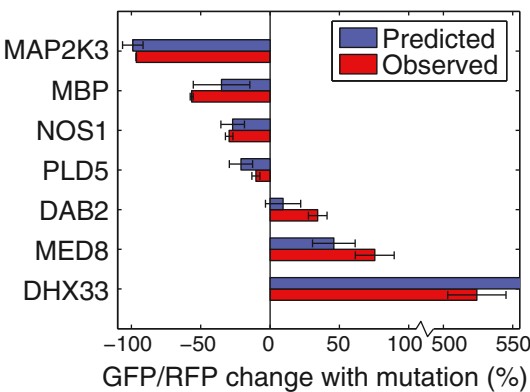

**Figure 7.  Effect of mutations on TIS reporter.**
TIS sequences from Table 1 (both with and without mutations) were inserted into the TIS reporter. The TIS sequences were analyzed individually by conventional flow cytometry, and the change in TIS efficiency caused by the TIS mutation was calculated. The observed changes in TIS reporter efficiencies were compared to the dinucleotide PWM predictions. Error bars represent 95% CI (n = 3). The predicted change in TIS efficiency for DHX33 was 798% (off-scale). Source data are available online for this figure.

truncated protein may have an altered biological function or, because many localization signals reside on the N-terminus, may have an altered cellular localization (Danpure, 1995). The effect of leaky scanning on proteomic diversity remains poorly investigated in part due to the difficulty in identifying when N-terminal truncated protein isoforms are translated (Bazykin & Kochetov, 2011).

To identify N-terminal truncated protein isoforms, we developed a quantitative leaky scanning model that incorporated our TIS efficiency values to predict the relative translation of truncated ORFs. To model the leaky scanning mechanism, we considered the sequence of two TISs: the annotated TIS and the nearest downstream TIS with an in-frame AUG start codon (Supplementary Fig S6). The ratio of the initiation occurring at the second TIS compared to the initiation occurring at the first TIS was described by

$$X = \frac{(100 - kE_1)}{100 E_2}$$

The parameters $E_1$ and $E_2$ are the relative efficiency values determined by the first and second TIS sequences, respectively. The constant $k$ relates the relative efficiency values to an absolute probability of initiation. We previously estimated $k$ to be 0.86 by analyzing the effect of synthetic uORFs on a downstream initiation reporter (Ferreira *et al*, 2013). Using the quantitative leaky scanning model, we predicted 1,023 human genes to have secondary protein isoforms where $0.50 \leq X < 1.0$ and 318 genes where $X \geq 1.0$ (Supplementary Table S4). In other words, for all 1,341 genes, we predict initiation to occur at a truncation TIS (i.e., initiation site of a truncated ORF) with a frequency greater than or equal to 50% of that of the annotated TIS. The predictions from our model suggest that 'leaky' ribosome scanning broadly increases the proteomic diversity.

We sought to corroborate the quantitative leaky scanning model predictions with previously reported ribosome footprint profiling data. Briefly, ribosome footprint profiling uses the small molecule drug harringtonine to prevent newly initiated ribosomes from translating, resulting in the accumulation of ribosomes at functional initiation

sites and the depletion of downstream ribosomes (Ingolia *et al*, 2009, 2011, 2012). By sequencing and aligning the protected mRNA fragments, one can determine the locations of translation initiation. A drawback of the method is that the harringtonine treatment skews initiation toward upstream TISs, since a ribosome arrested at an upstream TIS would block another scanning ribosome, one that has not yet reached the upstream TIS, from reaching the downstream TIS. In contrast, a ribosome arrested at the downstream TIS would not prevent a second ribosome from initiating at the upstream TIS.

We compared the quantitative leaky scanning model predictions (applied to the mouse transcriptome; Supplementary Table S5) to ribosome footprint profiling data from mouse embryonic stem cells (Ingolia *et al*, 2011). We filtered the data such that we only analyzed transcripts with a sufficient number of ribosome footprint reads ($\geq$ 50 at the annotated TIS), where the TISs were sufficiently separated ($\geq$ 20 bases), and no annotated alternative 5′ transcript isoforms existed. The 980 transcripts that fit our criteria were scored for evidence of initiation at the putative truncation TIS (Fig 8A). We observed evidence of a truncation TIS in 93% of the transcripts that had a predicted initiation ratio, $X \geq 1.0$. These results were compared to qualitative predictions based on −3/+4 position 'rules of thumb', which would have generated the most reasonable predictions prior to this work (Fig 8B). Transcripts where the annotated TIS contained a −3U and lacked a +4G would previously have been considered the most inefficient TIS sequences and therefore the most likely TISs to allow for leaky scanning (Kozak, 1995). We observed evidence of a truncation TIS in 63% of the transcripts that met these criteria. Therefore, the quantitative leaky scanning model was more accurate than simple qualitative −3/+4 position rules at predicting which genes were most likely to have an active truncation TIS ($P = 0.013$).

*Sap18*, *Npepps*, and *Hagh* are examples where the quantitative leaky scanning model was in strong agreement with the ribosome footprinting data. *Sap18*, which encodes for a histone deacetylase protein (McCallum *et al*, 2006), has a low-efficiency annotated TIS sequence UAGCUC<u>AUG</u>CU followed by the high-efficiency TIS sequence AGGAAG<u>AUG</u>GC (Fig 8C). The quantitative leaky scanning model predicted significant amounts of initiation at the truncation TIS. In support of this prediction, the ribosomal footprint profiling data showed peaks at both TISs, suggesting that initiation was occurring at both TISs. As a second example, *Npepps*, which encodes for an aminopeptidase protein (Yanagi *et al*, 2009), also demonstrated evidence of initiation at a truncation TIS (Fig 8D). The low-efficiency annotated TIS CGGUGA<u>AUG</u>UG was followed by the high-efficiency TIS sequence GCCGCG<u>AUG</u>CC. The quantitative leaky model predicted approximately equal initiation at the annotated TIS and at the truncation TIS, which was in agreement with ribosome footprint profiling data. A third example is *Hagh*, which encodes a glyoxalase II enzyme involved in metabolism (Fig 8E). The gene's annotated TIS sequence, UGGGUC<u>AUG</u>GU contains a −3G/+4G, which would traditionally be considered highly efficient (Harte *et al*, 2012). However, using the full TIS sequence, our quantitative leaky scanning model predicted that a considerable fraction of the ribosomes would leak past the annotated TIS and initiate at the truncation TIS (GACAUA<u>AUG</u>AA). The ribosome footprint profiling peaks at both the annotated TIS and the putative truncation TIS suggest that initiation occurs at both sites, supporting our predictive model. Overall, we found the quantitative leaky scanning model generally agreed with the ribosome footprint profiling data,

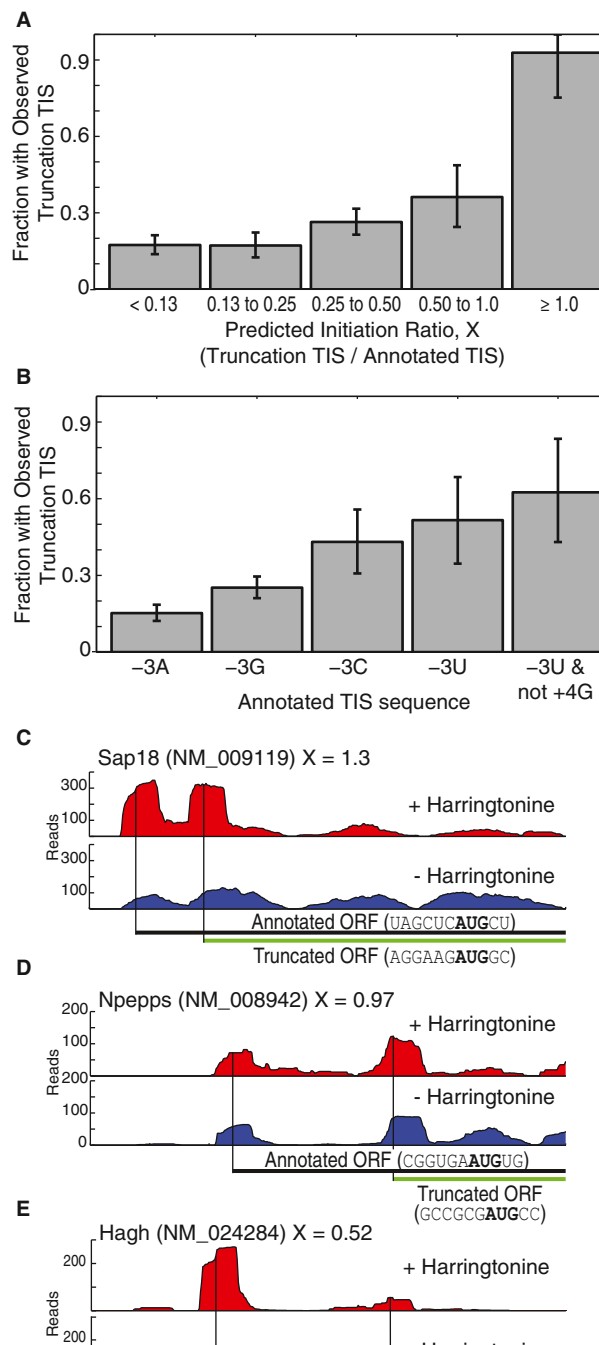

**Figure 8. Ribosome footprint profiling of genes with predicted truncation isoforms.**

*An in silico* quantitative leaky scanning model predicted the ratio, *X*, of initiation at truncation TISs relative to the initiation at annotated TISs. The predictions were compared to ribosome footprint profiling data from mouse embryonic stem cells.

A   The fraction of transcripts where ribosome profiling supports initiation at the truncation TIS grouped by predicted initiation ratio, *X*. Error bars represent 95% C.I.

B   Same as (A), but transcripts were grouped by the −3 and +4 positions of the annotated TIS.

C–E   Examples of transcripts with observed initiation at the truncation TIS. Ribosome density provided for transcripts with and without harringtonine treatment. The full-length annotated ORFs and the predicted truncated ORFs are indicated with their TIS sequences. Gene names, transcript names, and predicted initiation ratios are provided.

2012). Any quantitative prediction would need to account for variation in the efficiency of translation initiation, which in turn depends on the start codon context. As a step toward this goal, we employed FACS-seq to quantitatively measure the efficiency of translation initiation for a library consisting of 65,536 TIS sequences, which covers more TIS sequences than are present in the human genome. Using a dinucleotide PWM model to analyze this dataset revealed the importance of cooperativity between pairs of bases. The model also allowed us to process the FACS-seq raw data and more reliably gauge initiation efficiency. We generalized our findings with the high-efficiency motif RYMRMVAUGGC. Consistent with previous research (Kozak, 1986, 1987a), we found the −3 and the +4 positions to be important in determining TIS efficiency. However, our analysis also found the −2, −4, and +5 positions to be influential.

The divergence of consensus sequences within eukaryotic species has been used to argue that the initiation machinery, and its preference for TIS sequences, has undergone substantial evolution (Hinnebusch, 2011; Zur & Tuller, 2013). Indeed, there are differences between the reported consensus sequences for yeast (AMAMAAUGUCU (Cigan & Donahue, 1987; Hamilton *et al*, 1987)), higher plants (CAAMAAUGGCG (Joshi *et al*, 1997)), invertebrates (CAARAUGG (Pesole *et al*, 2000)), and vertebrates (GCCRCCAUGG (Kozak, 1987b)). However, the TIS usage does not necessarily equate to the ribosome's TIS sequence preference (Lukaszewicz *et al*, 2000). Except for the −5A and +4U in the yeast sequence, the three consensus sequences are remarkably similar to the high-efficiency mammalian TIS motif RYMRMVAUGGC, which we determined without incorporating any genome information. The differences in consensus sequences may be related to factors other than ribosome binding preferences. For example, eukaryotes with low GC-content genomes have been shown to favor A-rich TIS sequences (Nakagawa *et al*, 2008). In the case of yeast, which has a low GC-content genome, the scanning ribosome is particularly sensitive to secondary mRNA structure (Cigan & Donahue, 1987; Baim & Sherman, 1988). The high frequency of As in the TIS may reflect a selective pressure to prevent stable secondary structures while also maintaining high translational efficiency. In our study, we do not observe any appreciable relationship between secondary structure and TIS efficiency, perhaps because the mammalian eIF4 RNA helicase prevents stable secondary structures from interfering with the scanning ribosome. We hypothesize that much of the divergence in eukaryotic TIS consensus sequences could be due

demonstrating the ability of our model to predict the expression of translational protein isoforms from the mRNA sequence alone.

## Discussion

A long-term goal of genomics has been to use the genome sequence to predict proteome-wide expression (McLeay *et al*,

to variations in sensitivity of the preinitiation complex to RNA secondary structure near the TIS.

The debate surrounding the role, if any, of the +5 position originally inspired us to vary the +5 position in our TIS library (Boeck & Kolakofsky, 1994; Grunert & Jackson, 1994; Kozak, 1997; Xia, 2007; Nakagawa *et al*, 2008). We found that the +5 position had a large impact on TIS efficiency and displayed significant cooperativity with the +4 position. We cannot discount the possibility that the amino acid encoded by the +4 and +5 nucleotides affected our fluorescence reporter, either by altering the fluorescence per GFP molecule or by altering the GFP half-life. Since GFP fluorescence is generally robust to fusion proteins (Tsien, 1998), it is unlikely that a single amino acid fused to GFP altered the fluorescence per molecule. Alternatively, it is possible that the GFP half-life was affected by the phenomenon commonly referred to as the N-end rule. This rule states that the N-terminal amino acid affects protein stability (Hwang *et al*, 2010; Varshavsky, 2011). Since all TIS sequences in our library shared a +6C, according to the N-end rule, any TIS sequence without a +4G/+5G (Gly) or +4C/+5C (Pro) should have produced a GFP targeted for acetylation and eventual degradation. Therefore, if the N-end rule had significantly impacted our data, then TIS sequences with +4G/+5G or +4C/+5C should have produced a GFP with a longer half-life, increasing the GFP/RFP measurements. We did not observe this trend and, in fact, TIS sequences with +4C/+5C had some of the lowest GFP/RFP measurements (Fig 5A). Thus, our data suggest that the N-end rule had a minor effect, if any, on our GFP/RFP measurements.

The biophysical details of how the TIS sequence affects the ribosomal preinitiation complex remain unclear. Recently, Lomakin and Steitz (2013) determined the crystal structure of the mammalian preinitiation complex (48S PIC). The structure confirmed that the −6 to the +7 positions were within the ribosome's mRNA channel, allowing for the possibility that the +6 and +7 positions could also influence initiation. Downstream of the start codon, the +4 and +5 positions were proximal to 18S helix h44, eIF1, and the basic loop of eIF1A. Upstream of the start codon, the ribosomal proteins rpS26e and rpS28e were proximal to the influential −4, −3, and −2 positions. Additionally, the −3 position has been shown to UV-cross-link with eIF2α, which was not included in the crystal structure (Pisarev *et al*, 2006). Interestingly, the suboptimal −3U was cross-linked less efficiently with eIF2α than a −3G. It is possible −2 and −4 positions influence initiation by promoting the −3 interaction with eIF2α. Unfortunately, the preinitiation complex crystal structure resolution was too low to determine the orientation of the mRNA bases or the orientation of the residue side chains in the mRNA channel. A higher resolution crystal structure would enable us to determine not just which parts of the preinitiation complex are proximal to the TIS, but also the nature of these interactions, enabling a biophysical explanation for the TIS efficiency trends described here.

The comprehensive and quantitative analysis of TIS efficiency enabled us to perform a genome-wide analysis of TIS efficiency, to identify TIS mutations that could drive tumorigenesis, and to identify transcripts that encode truncated protein isoforms. Using a quantitative leaky scanning model that accounted for TIS efficiency, we predicted the occurrence of protein isoforms resulting from internal initiation within annotated ORFs, effectively generating truncations. The agreement between our model and the available ribosome footprint profiling data suggests that initiation events

previously attributed to internal ribosome entry site (IRES) structures may instead be a result of leaky scanning (Ingolia *et al*, 2011). For example, a truncation TIS in the *HAGH* gene had been experimentally validated (Cordell *et al*, 2004). Since the annotated TIS contained a −3G/+4G, which was thought to cause efficient initiation with minimal leaky scanning, the initiation at the truncation TIS was attributed to a putative IRES structure. However, our leaky scanning model, which considered the full TIS sequence (UGGGUCAUGGU), predicted substantial leaky scanning (predicted initiation ratio $X = 0.52$) (Fig 8E). While we cannot rule out the presence of an IRES, our results suggested that leaky scanning can account for some, if not all, of the initiation at the downstream TIS. Thus, our results agree with previous studies calling into doubt many of the putative mammalian IRES structures (Han & Zhang, 2002; Kozak, 2003, 2007; Bert *et al*, 2006; Elango *et al*, 2006).

By employing FACS-seq, a method combining FACS and high-throughput DNA sequencing, we were able to analyze a library of genetic reporters and determine the translation initiation efficiency for all possible TIS sequences utilizing an AUG start codon. The method is similar in design to other recent studies (Kinney *et al*, 2010; Sharon *et al*, 2012; Kosuri *et al*, 2013). To our knowledge, our implementation of FACS-seq has to date analyzed the largest number of sequences in one experiment (65,536 vs. 14,234 (Goodman *et al*, 2013)). With some optimization, we believe that FACS-seq can be applied to libraries approaching $10^6$ sequences, with the practical limit being determined by the time required to sort the library. Our ability to analyze such a large number of sequences was dependent on the precision of the readout from our translation reporter. As long as an effective fluorescent reporter can be designed, FACS-seq and similar methods should in principle enable comprehensive analysis of any DNA or RNA sequence library. Because the sequence space of biological motifs grows exponentially larger with each additional base, comprehensive analysis with traditional techniques can be challenging or impractical. Instead, massively parallel methods like FACS-seq enable the thorough analysis of a desired sequence space such that one can now predict the behavior of all sequence elements employed in the genome.

## Materials and Methods

### TIS reporter library construction

The TIS reporter library was constructed using degenerate primers followed by a high-efficiency Gibson reaction. Monomeric enhanced GFP (EGFP A207K, here referred to as GFP) was PCR amplified and gel purified using the forward primer 5′-CATCCTCTAGACTGCCGGATCTCGAGTAACTGACTAGT NNNNNN ATG NN CGAATTCAGCAAGGGCGAGGAG-3′ and reverse primer 5′-CGGAATTGGCCGCCCTAGATG-3′. The eight nucleotides that were varied are indicated by Ns in the degenerate forward primer. The three upstream stop codons in all three reading frames prevented an upstream start codon from translating GFP. To construct the plasmid library, pCru5-IRES-mCherry-F2A-Puro plasmid was digested with XhoI and NotI, gel purified, and then used along with the amplified GFP in an optimized Gibson reaction. TOP10 *E. coli* cells were transformed with the Gibson reaction product to yield greater than $10^6$ *E. coli* transformants. This yielded a > 99% chance of each possible

TIS sequence being present in at least one of the transformants (assuming that the reaction was not biased for specific TIS sequences). The colonies were scraped into a single flask followed by a DNA HiPure Maxiprep purification (Life Technologies, Carlsbad, CA).

## Cell culture

Retroviral particles of the TIS reporter library were produced by transiently transfecting HEK-293T cells with equal amounts of pCru5-GFP-IRES-mCherry-F2A-Puro DNA and pCL-Eco retrovirus packaging vector. The transfection followed the CalPhos Mammalian Transfection Kit protocol (Clonetech Laboratories, Inc., Mountain View, CA). Viral particles were harvested and filtered through a 0.4-μm filter.

FACS-seq experiments were carried out in PD-31 cells, an Abelson murine leukemia virus-transformed pre-B cell line. The cells were cultured in RPMI-1640 medium (Life Technologies) supplemented with 10% fetal bovine serum (FBS, Gemini Bio Products, Sacramento, CA), 2 mM glutamine, 1 mM sodium pyruvate, 0.05 mM 2-mercaptoethanol, 100 U/ml penicillin, and 100 μg/ml streptomycin at 37°C with 5% $CO_2$. To insure full library coverage, virus was added to $350 \times 10^6$ PD-31 cells with 8 μg/ml polybrene (hexadimethrine bromide). The infection frequency was 2.3% as judged by flow cytometry 2 days post-infection, insuring that transduced cells received a single copy of the vector. Puromycin (2 μg/ml) was added to the media 3 days post-infection to select for infected cells.

Additional experiments were carried out in a variety of cell lines and conditions. NIH-3T3 cells, HeLa cells, and HepG2 cells were cultured in DMEM medium (Life Technologies) with 10% FBS, 4.5 g/ml glucose, and 2 mM glutamine. PD-31 cells (standard conditions) and K562 cells were cultured in RPMI-1640 medium with 10% FBS, 2 mM glutamine, 1 mM sodium pyruvate, and 0.05 mM 2-mercaptoethanol. HCT-116 cells were cultured in McCoy's 5A medium (HyClone Laboratories, Logan, UT) with 10% FBS. Where indicated, 1.75 μM imatinib (Novartis, Basel, Switzerland) was added (Muljo & Schlissel, 2003; Ferreira & Wang, 2013). All cells were cultured with 100 U/ml penicillin and 100 μg/ml streptomycin at 37°C with 5% $CO_2$.

## Cell sorting

PD-31 cells transduced with the TIS reporter library were sorted based on the ratio of GFP to mCherry (RFP). All sorting was performed on an Aria II (BD Biosciences, Franklin Lakes, NJ) using the GFP channel (488 nm excitation laser, 505 nm splitter, 525/50 nm emission filter) and the mCherry channel (561 nm excitation laser, no splitter, 582/18 nm emission filter). The voltages of the GFP and RFP channels were reduced such that the GFP and RFP values of non-fluorescent cells were approximately 50 a.u. Cells were gated for RFP values greater than $10^3$ (i.e., 20× autofluorescence). Using the FACSDiva software (BD Biosciences), a GFP to RFP ratio parameter (here referred to as GFP/RFP) was created. On the day of sorting, a preliminary reading was taken of the TIS library expressing cells. Custom MATLAB software (MathWorks, Natick, MA) analyzed the preliminary data to calculate the correct gate spacing so that 5% of the cells fell into each of the 20 gates. $10^6$ PD-31 cells were sorted into each of the 20 gates.

## Barcoding and sequencing

The TIS sequences from each sorted population were PCR amplified, barcoded, and sequenced. The PD-31 cells were cultured for 2 days after cell sorting. Genomic DNA (gDNA) was isolated with DNeasy spin column (Qiagen, Venlo, Netherlands). An isopropanol precipitation was used to concentrate the gDNA. A mean of 22 μg of gDNA was added to each of the 20 PCRs using barcoded primers (Supplementary Table S6). Each barcoding PCR was 100 μl in size and used OneTaq HS with the standard buffer (New England Biolabs, Ipswich, MA). The PCR products were gel extracted, purified, and mixed in equimolar ratios. A second PCR was used to add the standard Illumina sequencing adapters (Supplementary Table S6). The final PCR product was sequenced with the Illumina Genome Analyzer IIx (San Diego, CA).

## Data analysis

The TIS sequences and barcodes were extracted from the DNA sequencing reads to create the FACS-seq histograms. The sequences were first normalized by barcode number, such that each barcode represented an equal fraction of the total population. The gate coordinates used for the cell sorting were used to simulate the original GFP/RFP histograms and to calculate the median translation initiation (GFP/RFP). All of the analysis was performed with custom MATLAB scripts.

## Position weight matrices

To generate a position weight matrix, linear regression was performed using the natural logarithm of the raw FACS-seq data as the dependent variable and the TIS sequence as the independent variables (Barrick *et al*, 1994; Salis *et al*, 2009). By using the logarithm, we were modeling the interaction between the mRNA and the ribosomal preinitiation complex as an association reaction at equilibrium. Hence, the efficiency of translation initiation, *E*, would be described by:

$$E \propto \exp(-\Delta G/RT)$$

where $\Delta G$ is the change in Gibbs free energy upon binding of the preinitiation complex with the TIS sequence, *R* is the ideal gas constant, and *T* is temperature. Using this rational, the mononucleotide PWM modeled the efficiency as:

$$E = k \cdot \exp\left(\sum_{i=-6}^{+5} C_{b,i}\right)$$

where *k* was a proportionality constant and $C_{b,i}$ was the coefficient for base *b* = {U, C, A, or G} at position *i* = {−6, −5, −4, −3, −2, −1, +4, or +5}. The coefficients were normalized such that:

$$\sum_b \frac{\exp(C_{b,i})}{4} = 1$$

which caused the coefficients to be proportional to the specific free energies of ribosome binding (Barrick *et al*, 1994).

The dinucleotide PWM was constructed in a manner similar to the mononucleotide PWM except the independent variables included all possible pair-wise interaction.

$$E = k \cdot \exp\left(\sum_{i=-6}^{+5} C_{b,i} + \sum_{i=-6}^{+5} C_{b_1,i_1,b_2,i_2}\right)$$

where $C_{b,i}$ matched the mononucleotide PWM coefficients, and $C_{b_1,i_1,b_2,i_2}$ was the interaction coefficient for base $b_1$ at position $i_1$ and base $b_2$ at position $i_2$ (subject to $i_1 \neq i_2$). The coefficients were normalized such that

$$\sum_{b_1 b_2} \frac{\exp(C_{b_1,i_1,b_2,i_2})}{16} = 1$$

The dinucleotide PWM was applied using custom MATLAB and R scripts. For genome-wide predictions, the mouse and human RefSeq GenBank files (release 55, September 2012) were downloaded from NCBI (http://www.ncbi.nlm.nih.gov/refseq/). The AUG-containing TIS sequences were extracted from the mRNA sequence and used to predict TIS efficiency values.

## mRNA secondary structure

All mRNA secondary structure predictions were performed with the open source NUPACK software (Zadeh *et al*, 2011). The mRNA folding free energy ($\Delta G$) was calculated using the pfunc command. The temperature was set at 37°C, dangle energies were included (-dangles all), and the default RNA parameters were used (-material rna1995).

## SNP analysis

Cancer-related mutations were identified by intersecting COSMIC mutations (version 65 full export updated May 28, 2013) with the annotated human TIS sequences (GRCh7/hg19) using bedTools (Quinlan & Hall, 2010). Mutations in these regions were evaluated using the dinucleotide PWM to predict the change in expression. Genes shown in Table 1 were evaluated for expression and copy number changes using publicly available data from The Cancer Genome Atlas Project using the Cancer Genome Browser (Goldman *et al*, 2012) and the Copy Number Portal. All statistical analyses on expression data were performed using a Student's *t*-test with a Bonferroni multiple hypothesis testing correction. Copy number variation *P*-values came from the Copy Number Portal.

## Ribosome footprint profiling analysis

Ribosome footprint profiling datasets from Ingolia *et al*, 2011, were downloaded from NCBI SRA browser (http://www.ncbi.nlm.nih.gov/sra): SRR315612, SRR315613, SRR315614, SRR315615 (mESC treated with 1 μg/ml harringtonine followed by 100 μg/ml cyclohexamide) and SRR315602, SRR315601 (mESC treated with 100 μg/ml cyclohexamide). After removing low-quality reads and the cloning adapter, ribosomal RNAs were removed using Bowtie. The remaining reads were mapped to the mouse genome (NCBI37/mm9).

Scoring of ribosome footprint profiles was performed using a custom MATLAB graphical user interface. The user was displayed a trace of the ribosome footprint data with the location of the annotated TIS and the putative truncation TIS. The user was asked to manually score if the ribosome footprint data supported the putative truncation TIS. As with all visual scoring methods, human bias was a major concern. In order to limit any bias, the transcripts were scored in a random order and with no additional information provided to the user. An attempt was made to automate the TIS peak calling. However, the results were inconsistent with visual inspection and were, in our opinion, unsatisfactory. Because we had a limited number of transcripts to score, we chose a manual process that emphasized accuracy over throughput. For full transparency, all transcript traces, predictions, and scoring are provided in Supplementary Fig S7.

**Supplementary information for** this article is available online: http://msb.embopress.org

## Acknowledgements

We thank Joshua Elias (Department of Chemical and Systems Biology, Stanford University) and the entire Wang laboratory for input and stimulating discussions. This study was supported by the NIH (5R21AG040360-02). WLN was supported by the Department of Energy Office of Science Graduate Fellowship Program (DOE SCGF), made possible in part by the American Recovery and Reinvestment Act of 2009, and administered by ORISE-ORAU under contract no. DE-AC05-06OR23100. WLN was also supported by Stanford Bio-X Graduate Student Fellowship.

## Author contributions

WLN designed and performed experiments, analyzed data, and prepared the manuscript; RJF and JZ processed ribosome footprint profiling data; AB performed bioinformatics related to TIS mutations; AJD analyzed data and scored ribosome footprints; PAK supervised RJF and AB; CLW conceived of FACS-seq, performed preliminary experiments, and directed the project.

## Conflict of interest

The authors declare that they have no conflict of interest.

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
