## [Review Process File · Molecular Systems Biology]

Quantitative analysis of mammalian translation initiation sites by FACS-seq

William L. Noderer, Ross J. Flockhart, Aparna Bhaduri, Alexander J. Diaz de Arce, Jiajing Zhang, Paul A. Khavari, Clifford L. Wang

Corresponding author: Clifford L. Wang, Stanford University

Review timeline:

Submission date:	17 January 2014
Editorial Decision:	27 February 2014
Revision received:	21 May 2014
Editorial Decision:	30 June 2014
Revision received:	22 July 2014
Accepted:	24 July 2014

Editor: Maria Polychronidou

Transaction Report:

1st Editorial Decision

27 February 2014

Thank you again for submitting your work to Molecular Systems Biology. We have now heard back from two of the three referees whom we asked to evaluate your manuscript. Since their recommendations are overall similar, I prefer to make a decision now rather than further delaying the process. As you will see from the reports below, the referees acknowledge that the presented findings are potentially interesting. However, they raise a series of significant concerns, which should be carefully addressed in a revision of the manuscript.

Overall, the referees point out that additional analyses are required to more convincingly support the main findings. Without repeating all the points listed below, some of the more fundamental issues are the following:

- A subset of the library needs to be examined under different conditions (i.e. using a different reporter gene, different cell line, different growth conditions), to exclude that the observed effects on translation initiation efficiency are context-dependent.
 - Additional analyses are required in order to provide mechanistic insights into how the different Kozak sequences influence translation efficiency. In particular, as reviewer #3 points out, the role of mRNA structure needs to be taken into consideration and analyzed.
 - Statistical analyses need to be carefully performed throughout the study.
-

Reviewer #1:

Noderer et al. performed a large scale study to determine the efficiency of translation initiation sites (TIS) surrounding the AUG START codon, based among others, on synthetic biology experiments combining fluorescence-activated cell sorting with high-throughput DNA sequencing. They

measured the translation rates from a library representing all 65,536 possible sequence motifs spanning the -6 to +5 positions. Amongst others they show that the motif RYMRMVAUGGC is a translation initiation enhancer; and that the best model of translation initiation should be based on a PWM of nucleotide pair-wise interactions. In addition, they connect their model to somatic TIS mutations associated with tumorigenesis and used it to predict alternative initiation sites. The study is interesting, but as described below the analysis and presentation should be improved; furthermore, additional experiments will enhance the paper.

Major

1) The authors ignore many very relevant previous studies in the field. These studies should be cited, and the current result should be discussed in light of them. The following is a partial list: Goodman DB, Church GM, Kosuri S (2013) Causes and effects of N-terminal codon bias in bacterial genes. *Science* 342: 475-479. doi: 410.1126/science.1241934. Epub 1242013 Sep 1241926.

Plant Mol Biol. 1997 Dec;35(6):993-1001. Context sequences of translation initiation codon in plants. Joshi CP, Zhou H, Huang X, Chiang VL.

PLoS Comput Biol. 2013;9(7):e1003136. doi: 10.1371/journal.pcbi.1003136. Epub 2013 Jul 11. New universal rules of eukaryotic translation initiation fidelity. Zur H, Tuller T.

Nakagawa S, Niimura Y, Gojobori T, Tanaka H, Miura K (2008) Diversity of preferred nucleotide sequences around the translation initiation codon in eukaryote genomes. *Nucleic Acids Res* 36: 861-871 Epub 2007 Dec 2007.

Gene. 2000 Dec 30;261(1):85-91. Analysis of oligonucleotide AUG start codon context in eukaryotic mRNAs. Pesole G, Gissi C, Grillo G, Licciulli F, Liuni S, Saccone C.

Mapping the translation initiation landscape of an *S. cerevisiae* gene using fluorescent proteins. Ben-Yehzekel T, Zur H, Marx T, Shapiro E, Tuller T. *Genomics.* 2013 Oct;102(4):419-29. doi: 10.1016/j.ygeno.2013.05.003. Epub 2013 May 28.

Lee S, Liu B, Huang SX, Shen B, Qian SB (2012) Global mapping of translation initiation sites in mammalian cells at single-nucleotide resolution. *Proceedings of the National Academy of Sciences* 109: E2424-E2432.

Danpure CJ (1995) How can the products of a single gene be localized to more than one intracellular compartment? *Trends Cell Biol* 5: 230-238.

Plant Sci. 2000 May 15;154(1):89-98. In vivo evaluation of the context sequence of the translation initiation codon in plants. Lukaszewicz M, Feuermann I M, J rouville B, Stas A, Boutry M.

2) One possible disadvantage of the approach is the fact that all the experiments are based on one (GFP) gene with sequence modifications near the beginning of the ORF. Thus, the results probably tend to model/describe translation initiation efficiency in this context (The GFP protein). Though the authors discuss this issue and show that at least part of the signal can be seen in endogenous genes, I think it will be very helpful to repeat the study with a different reporter gene (e.g. a YFP gene) and compare/correlate the results in both experiments. This will enable the reader to gain some intuition regarding the fraction of the signal which is specific only to the GFP system.

3) Please report p-values in all places. For example when you report the correlation with the models "(mononucleotide PWM, $R^2 = 0.52$; raw data, $R^2 = 0.44$)".

4) Comparison of the different PWM models: the number of parameters in the dinucleotide PWM model is much higher than the number of parameters in the mononucleotide PWM. Thus, it is not clear to me if the improvement gained by the dinucleotide model is not due to overfitting. One possible strategy to check this point is to perform cross validation. If the authors performed such a procedure it should be clearly explained (In the current version it was not clear if indeed this what was performed).

5) Another way to make the paper stronger is to perform the experiment in different conditions (e.g. various perturbations of the HEK cells or different cell cycle stages).

6) Page 7: "Next, we analyzed the initiation efficiency for all human protein-coding TISs... In summary, we found that protein-coding ORFs were enriched for more efficient TIS sequences, while uORFs were not." All the results mentioned in this section should be followed by a p-value; currently, it is hard to interpret them. In addition, the results are clearly not surprising (we expect to see that protein-coding ORFs were enriched for more efficient TIS sequences). It may be interesting (and relatively easy) to check if there is correlation between the translation initiation efficiency of human ORFs (based on the model inferred by the experiment) and their protein levels (or any other measure related to their expression levels).

7) The subsection "Somatic TIS mutations associated with tumorigenesis": I understand that the mutations introduced in the GFP protein? and are not based on the cancerous gene (or part of it) fused to the GFP? (the second possibility is of course "stronger")
In any case, this point should be better emphasized. It may be interesting/helpful to compute a p-value related to the effect of the mutations associated with tumorigenesis on translation initiation (based on their model) vs. the effect of the mutations in this region not associated with tumorigenesis on translation initiation.

8) Methods; subsection "Ribosome footprint profiling analysis": "Scoring of ribosome footprint profiles was performed using a custom MATLAB graphical user interface. The user was displayed a trace of the ribosome footprint data with the location of the annotated TIS and the putative truncation TIS. The user was asked to manually score if the ribosome footprint data supported the putative truncation TIS. As with all visual scoring methods, human bias was a major concern. In order to limit any bias, the transcripts were scored in a random order and with no additional information provided to the user. An attempt was made to automate the TIS peak calling. However, the results were inconsistent with visual inspection and were, in our opinion, unsatisfactory. Because we had a limited number of transcripts to score, we chose a manual process that emphasized accuracy over throughput. All transcript traces, predictions, and scoring are provided in Supplementary Figure 4." This part of the analysis should be changed and based on a computational/mathematical scoring (which should be clearly defined/described) followed by p-values; it is unacceptable to provide results based on manual human dependent analysis that cannot be reproduced in future studies.

9) The section "Quantitative leaky scanning model of alternative initiation". The comparison of the prediction of the translation initiation model to ribosomal profiling data is important; however, this section should be significantly improved. First, I suggest to also check the data of Lee et al. (see point 1)); second, all the genes should be analysed (the authors write "we chose to analyze 235 transcripts with predicted initiation ratio, $X \geq 0.33$ and 235 transcripts whose predicted initiation ratio, X , was < 0.33 "); third, also here the authors should provide quantitative measures based on all the ATG codons such as correlations, p-values and error-rates connecting their predictions to the ribosomal density measurements.
In addition, it is not clear to me why the authors did not analyze out of frame downstream AUG start codons and upstream codons, and compared their prediction to the measurements of the ribosomal profiling data.

Minor

1) Please add to the abstract quantitative measures such as correlations, error rates, relative translation, etc to support the reported conclusions.

2) Subsection "Position Weight Matrices" of the Methods section: I understood the formulas but they can be improved and be written in a more clear/accurate manner. For example, the score of a sequence with nt $b=-6:+5$ is $k \cdot \exp(\sum_b C_{b,i})$ where $C_{b,i}$ is the coefficient of nt i ($i = \{A, T, C \text{ or } G\}$)

Reviewer #3:

The authors combine next-generation sequencing with flow cytometry to measure the translation rate efficiency for 65,536 Kozak sequence motifs controlling start codon selection in a mammalian cell line. Using this data, they parameterize a di-nucleotide nearest-neighbor model of translation rate efficiency, defined as the fraction of time that a start codon's open reading frame is selected for

translation. The authors apply the model to predict the translation efficiency for protein coding sequences in the human genome. They find a small number of somatic cell mutations in Kozak sequences that are related to tumor formation, and a small number of ORFs with leaky Kozak sequences that result in expression of truncated protein isoforms.

Overall, this manuscript describes results that are interesting and potentially useful towards relating genome sequence mutations to changes in protein expression. However, the current version of the manuscript is very light on mechanistic detail, and analyzes the relationship between Kozak sequence and translation rate efficiency using only informatics. The authors focus solely on the Kozak sequence and find relatively small differences in translation rate efficiency (~7-fold, based on Figure 4) when mutating the Kozak sequence, but clearly protein expression levels in mammals vary more than 7-fold. The data analysis could also be more quantitative, and should account for measurement variation when comparing the relatively small differences in translation efficiency between Kozak sequences. The analysis of the effects of somatic cell mutations also requires further replication measurements and careful statistical analysis to ensure that the differences between predictions and measurements are small enough to be statistically significant. There are also several key details that should be described up-front in the main text.

Specific comments:

1. The authors need to introduce the eukaryotic translation process with more mechanistic detail, and provide several examples from the literature to show how changes to the 5' UTR sequence, besides the Kozak sequence, can greatly alter translation rate. The manuscript should clearly indicate that the number of ribosomes that bind to a eukaryotic mRNA is not determined by the Kozak sequence, but is instead controlled by mRNA sequences and structures farther upstream as well the cap-bound protein complex that mediates 43S binding. The Kozak sequence determines the fraction of scanning ribosomes that select a start codon for translation initiation. This will explain why two mRNAs can have the same Kozak sequence, but in fact can express very different amounts of protein.

The terminology in the article "The initiation of mammalian protein synthesis and mRNA scanning mechanism" by Lomakin & Steitz should be used to describe 43S ternary complex formation, 43S scanning, and formation of the 48S PIC when pausing in front of the Kozak occurs.
<http://www.nature.com/nature/journal/vaop/ncurrent/full/nature12355.html>

2. Related to #1, the authors must describe the sequence and structure of their 5' UTR because it is essential towards comprehensive analysis and use of their model. This description must include calculations to indicate the presence of stable RNA structures in the 5' UTR, their location, and proximity to 5' methylG cap and to the Kozak sequence.

3. The measurement precision of the FACS-Seq approach is not adequately addressed. First, the distribution of TIS sequence reads in each of the 20 sorted bins is non-Gaussian. Some distributions have very long tails, indicating that the same TIS sequence can cause large changes in translation efficiency, due to some confounding or time-dependent variable. The authors do not provide an explanation, and overly simplify the analysis by only using the median of the distribution to quantify the overall translation efficiency for each TIS sequence. The authors should calculate the variance of the distributions and show via a two-tailed significance test that TIS sequences with a median translation rate of X have statistically different read sequence distributions, compared to TIS sequences with a median translation rate of X+dX, where dX is the maximum precision that could be claimed for the FACS-Seq approach. The authors should then take care to avoid making comparisons where the difference is less than dX as those will not be statistically significant comparisons. When parameterizing models with FACS-Seq data, or comparing model predictions to FACS-Seq measurements, one should not expect a greater accuracy than the FACS-Seq precision unless by fortuitous chance. The authors supplement their FACS-Seq measurements using (more precise) flow cytometry measurements of individual TIS sequences; the same statistical significance testing must be performed for these measurements.

4. The maximum range of translation rate efficiencies when mutating the Kozak sequence is never reported, and must be reported. Instead, the translation rate efficiencies are quantified in terms of %s, which can be misleading. Based on Figure 4, it appears that the maximum change in translation

rate efficiency by a mutation in the Kozak sequence will be about 7-fold. However, the reader should not have to guess-imate when reading the manuscript.

5. Two-tailed significance tests should be performed on the comparisons between the PWM models and flow cytometry measurements, shown in Figure 4. By themselves, R^2 values can be misleading. In this case, the conclusion should not be different.

6. A key result from the FACS-Seq measurements is that the consensus Kozak sequence is RYMRMVAUGGC. However, a gaping unanswered question is Why? Is there a mechanistic reason why this particular Kozak sequence, which is longer and more specific than previously proposed ones, maximizes the translation rate efficiency? Keeping in mind that the FACS-Seq measurements were performed in only one cell line, under one growth condition, using an existing upstream 5' UTR sequence that was not perturbed, and using one set of fluorescent protein reporters, the question of why becomes more important as there could be non-observed, unaccounted changes in a confounding variable that becomes the true reason for the extended consensus sequence. If there was corroborating evidence from the literature that this particular Kozak sequence has stronger interactions with the 43S ribosome, then it would answer these questions suitably. Otherwise, the additional "important" nucleotides might only be important in this specific 5' UTR, and not others in the human genome.

Figures 2 and 3 in the article by Lomakin & Steitz show the 43S (scanning) and 48S (paused) ribosomes in complex with mRNA, including the Kozak: Figure 2b shows a crystal structure of the 43S ribosomal complex with initiation factors, initiator tRNA, and mRNA bound together. The mRNA's start codon and surrounding sequences interact with specific ribosomal RNA helices. These interactions cause rotation of the ribosome's head domain (pausing), which then initiates 60S recruitment and translation elongation. Figure 3 shows a crystal structure of the 48S ribosomal complex in its paused state (after rotation) and the contacts with the Kozak sequence that were necessary to achieve head rotation. The section in Lomakin & Steitz's analysis, entitled "The mRNA channel and the mRNA path" discusses the role of the Kozak sequence in pausing. The authors should use the crystal structure to explain why nucleotide mutations from -6 to +7 alter the probability of pausing. They have the opportunity to use some molecular modeling to explain how these nucleotide mutations alter the interactions inside the binding pocket.

7. The authors neglect the role of mRNA structures that could affect start codon selection and translation efficiency. First, mutation of Kozak sequences could affect the formation of mRNA structures in distant locations, farther upstream of the Kozak. Upstream mRNA structures have been implicated in preventing 43S ribosome from binding the methylG cap, and pausing the 43S ribosome during scanning (PMC1440912). It is also possible that pausing of the 43S by a mRNA structure at the appropriate position over a start codon could actually increase start codon selection. The authors should analyze the compendium of mRNA structures that form in their 65,636 sequences and determine whether low or high translation rate efficiency was instead controlled by the presence or absence of mRNA structures upstream or downstream of the start codon.

8. The statistics in the section on genome-wide TIS predictions are not accompanied by any significance testing. If I flip 10 coins and receive 15% more heads than the average, that can happen by pure chance. If I flip 10000 coins, and receive the same result, that would be significant. Here, the results are closer to the latter, but the two-tailed p-value needs to be calculated. Because the distribution of translation rate efficiencies is likely non-Gaussian, it also helps to include a Kullback-Leibler Divergence plot in the Supp. Info to compare random to actual translation rate distributions.

9. In referring to Grave's disease, the sentence "The polymorphism was found to decrease the TIS efficiency by 16%, demonstrating that even modest changes ... can be relevant" cites only the in vitro result from Jacobson et. al. When they repeated CD40 expression measurements in vivo, they observed 32% less surface CD40 expression in fibroblasts, and 39% less surface CD40 surface expression in B cells. The difference between 16% and 32% is significant when the authors' follow-up analysis reports differences of 27% to 46% as biologically meaningful results, which may be false if one objectively considers the existing literature.

10. Figure 7 does not have error bars and no statistical significance testing has been performed. This

is particularly troubling since the changes in expression with vs. without the mutation are less than 50% for half of the TIS sequences, and the difference between model predictions and measurements appears to be greater than 20% for two of the TIS sequences. The usage of percentage as a unit further obscures the differences between the actual fluorescence data. The authors must perform these measurements in triplicate and carry out two-tailed significance tests.

11. There is a certain amount of tautology in the analysis of the small number (7) of somatic cell mutations that were analyzed. The authors "chose mutations where the predicted change in protein translation efficiency was consistent with known tumor expression patterns", and then concluded that "In summary, we have linked changes in TIS efficiency with known tumorigenic expression patterns". The authors can address this concern by expanding their analysis not only to SNPs found in TIS sequences related to tumorigenesis, but to TIS sequence SNPs that are associated with any disease condition. The authors should strive to present a sufficiently large number of validated predictions to convince the reviewers that the model does not have confounding variables. While a p -value < 0.05 is the standard bare minimum, it is not persuasive.

12. Regarding the leaky scanning model section, there is no explanation why 980 transcripts fit the authors' criteria, but only 470 transcripts were analyzed. The authors should analyze the 980 transcript data-set.

13. Overall, the section on the leaky scanning model repeats existing ribosome profiling data; using the PWM model does not explain any surprising observations or contribute to further understanding. It would be more interesting if low-efficiency nucleotides in the extended Kozak sequence, and not just the original Kozak sequence, were responsible for a truncation TIS that was biologically relevant.

14. The sequence space of biological motifs grows combinatorially, not exponentially, larger with each additional base. Exponentially is e^n . In this case, combinatorial is 4^n . $4 > e$.

15. The sentence "Our ability to analyze such a large number of sequences with relatively high resolution " is not supported by the absence of measuring the precision of the FACS-Seq approach or the quantitative reporting of the translation rate range that was observed.

16. The authors propose FACS-Seq as a technique for characterizing large sequence spaces. However, they do not provide any caveats or limitations to the FACS-Seq approach. Certainly, the authors would agree that FACS-Seq can not characterize 10^{12} sequences in a single-run, and yet this is the number of sequences in a 20 nucleotide motif. With these constraints, what are the most intelligent applications of FACS-Seq? Can one "predict the behavior of all sequence elements employed in the genome" using only FACS-Seq and informatic relationships?

17. The cell lines utilized need to be described in the main text. These are not only methodological details, but could affect the central conclusions of the manuscript.

18. It was also troubling that "linear regression was performed using the natural logarithm of the raw FACS-seq data as the dependent variable". In Salis et. al., the relationship between the ribosome's binding free energy to a mRNA was log-linear because, according to statistical thermodynamics, the probability of ribosome binding was proportional to $\exp(-dG/RT)$, where dG is the amount of reversible work needed to bind. Here, is there any expectation, based on the scanning mechanism, that the efficiency of start codon selection would be related to the logarithm of protein fluorescence? No, there should be a linear relationship between the probability of start codon selection and the resulting protein fluorescence. Because this is a purely informatic relationship, log-transforming protein fluorescence will simply alter the parameterized coefficients. The authors should repeat the parameterization using protein fluorescence as the dependent variable and receive other-valued coefficients. These coefficients would likely give you a similar PWM prediction. This is an example of why the numbers in informatic models have no physical or biological meaning.

May 20, 2014

We are submitting the revised paper entitled, “*Quantitative analysis of mammalian translation initiation sites*,” for your consideration. We were encouraged that the reviewers were enthusiastic about our work and appreciated their thoughtful comments. To address these concerns, we have revised the text, performed additional experiments, re-analyzed data, and added two supplementary figures. Here, we summarize the key questions raised by the reviewers and the revisions to the manuscript.

1. A subset of the library was examined under different conditions. We have measured the efficiency of individual TIS sequences in varied growth conditions, in multiple human and mouse cell lines, and using a different reporter gene. The observed TIS efficiencies were consistent, thus demonstrating that the results presented in this manuscript are not specific to our experimental system. We have added Supplementary Figure 2 to display these results and discussed the findings in the main text (Results: paragraph 10).

2. We have performed additional analysis on the mRNA secondary structure and its influence on initiation efficiency. We predicted the mRNA folding free energy for 65,536 mRNA sequences, each sequence consisting of a single TIS sequence and the 70 bases upstream and 70 bases downstream. These mRNA sequences represent mRNA folding near the initiation site for all possible TIS sequences. We observe no significant relationship between the predicted secondary structure free energy and the observed TIS efficiency (p-value = 0.18). We have added Supplementary Figure 3 to display these results and have discussed the findings in the main text (Results: paragraph 11; Supplementary Figure 3).

3. Statistical analysis has been improved. Where appropriate we have included further statistical analysis. None of the findings in this paper were altered due to the additional statistical analysis. We hope that the additional analysis has made our findings more convincing. Specifically, we have:

- Added p-values to all of our linear regressions. (Results: paragraph 6-7).
- Added 95% confidence intervals when changes in the average efficiency discussing dinucleotide interactions
- Created Kullback-Leibler Divergence plot (requested by reviewer #3) to compare the efficiency distributions from Figure 6 (Results: paragraph 13; Supplementary Figure 5).
- Added 95% confidence intervals when discussing the change in mean efficiency of the protein-coding TISs vs. TIS sequence space and the uORF TISs vs TIS sequence space (Results: paragraph 13).
- Performed the SNP experiment in triplicate and added error bars representing 95% confidence intervals to both the observed and predicted change in TIS efficiency (Figure 7).
- Analyzed an addition 490 ribosome footprint profiling traces (Figure 8, Results: paragraph 20)
- Added p-value in describing the difference in accuracy of the quantitative leaky scanning vs. qualitative -3/+4 rules in predicting truncation TISs (Results: paragraph 20)

Additionally, we have provided a detailed point-by-point response to every comment below. Thank you again for your efforts.

Detailed responses to the reviewers

(Reviewers comments italicized)

Response to reviewer #1:

1) *The authors ignore many very relevant previous studies in the field. These studies should be cited. and the current result should be discussed in light of them. The following is a partial list:*

Goodman DB, Church GM, Kosuri S (2013) Causes and effects of N-terminal codon bias in bacterial genes. Science 342: 475-479. doi: 410.1126/science.1241934. Epub 1242013 Sep 1241926.

Plant Mol Biol. 1997 Dec;35(6):993-1001. Context sequences of translation initiation codon in plants. Joshi CP, Zhou H, Huang X, Chiang VL.

PLoS Comput Biol. 2013;9(7):e1003136. doi: 10.1371/journal.pcbi.1003136. Epub 2013 Jul 11. New universal rules of eukaryotic translation initiation fidelity. Zur H, Tuller T.

Nakagawa S, Niimura Y, Gojobori T, Tanaka H, Miura K (2008) Diversity of preferred nucleotide sequences around the translation initiation codon in eukaryote genomes. Nucleic Acids Res 36: 861-871 Epub 2007 Dec 2017.

Gene. 2000 Dec 30;261(1):85-91. Analysis of oligonucleotide AUG start codon context in eukariotic mRNAs. Pesole G, Gissi C, Grillo G, Licciulli F, Liuni S, Saccone C.

Mapping the translation initiation landscape of an S. cerevisiae gene using fluorescent proteins. Ben-Yehzekel T, Zur H, Marx T, Shapiro E, Tuller T. Genomics. 2013 Oct;102(4):419-29. doi: 10.1016/j.ygeno.2013.05.003. Epub 2013 May 28.

Lee S, Liu B, Huang SX, Shen B, Qian SB (2012) Global mapping of translation initiation sites in mammalian cells at single-nucleotide resolution. Proceedings of the National Academy of Sciences 109: E2424-E2432.

Danpure CJ (1995) How can the products of a single gene be localized to more than one intracellular compartment? Trends Cell Biol 5: 230-238.

Plant Sci. 2000 May 15;154(1):89-98. In vivo evaluation of the context sequence of the translation initiation codon in plants. Lukaszewicz M, Feuermann1 M, Jérrouville B, Stas A, Boutry M.

We have cited all of the above studies, and have used them to enrich our discussion.

2) One possible disadvantage of the approach is the fact that all the experiments are based on one (GFP) gene with sequence modifications near the beginning of the ORF. Thus, the results probably tend to model/describe translation initiation efficiency in this context (The GFP protein). Though the authors discuss this issue and show that at least part of the signal can be seen in endogenous genes, I think it will be very helpful to repeat the study with a different reporter gene (e.g. a YFP gene) and compare/correlate the results in both experiments. This will enable the reader to gain some intuition regarding the fraction of the signal which is specific only to the GFP system.

This is an excellent point. In response to this comment, we created a TIS reporter that uses a YFP gene (mCitrine to be exact) in place of the mEGFP. The results from the two reporter constructs were in general agreement ($R^2 = 0.76$, $p = 1.1e-19$), demonstrating that our findings are not specific to GFP. We discuss these results in the text and have added supplementary figure (Results: paragraph 10; Supplementary Figure 2)

3) Please report p-values in all places. For example when you report the correlation with the models "(mononucleotide PWM, $R^2 = 0.52$; raw data, $R^2 = 0.44$)".

As requested, we added p-values to all of our linear regressions. In general we have increased the statistical rigor throughout the manuscript. See above for details on all of the additional statistical analysis.

4) Comparison of the different PWM models: the number of parameters in the dinucleotide PWM model is much higher than the number of parameters in the mononucleotide PWM. Thus, it is not clear to me if the improvement gained by the dinucleotide model is not due to overfitting. One possible strategy to check this point is to perform cross validation. If the authors performed such a procedure it should be clearly explained (In the current version it was not clear if indeed this what was performed).

Whenever a model is fit to experimental data, care must be taken to limit overfitting. One method to estimate the amount of overfitting is through cross-validation, as the reviewer suggests. Another method of estimating the amount of overfitting is to compare the model prediction to test data that was not used to train the model. A model that was overfit to the training data will perform poorly when validated with test data. We compared our PWM models, which were trained on the FACS-seq dataset, to data obtained from individual control constructs analyzed with traditional flow cytometry. In fact the tri-nucleotide PWM had a worse R^2 value than the di-nucleotide PWM. This we likely caused by two factors. (1) Tri-nucleotide interactions might not be as relevant as mono- and di-nucleotide interactions and (2) the tri-nucleotide PWM may be overfitting the FACS-seq data because the tri-nucleotide PWM has many more fitted parameters (i.e., increased model variance). We have tried to emphasize this point in the text (Results: paragraph 7).

5) Another way to make the paper stronger is to perform the experiment in different conditions (e.g. various perturbations of the HEK cells or different cell cycle stages).

We agree with the reviewer. We tested the TIS construct in 5 cell lines, in addition to PD31s. We observed consistent TIS efficiency measurements ($R^2 = 0.92$, $p\text{-value} < 2 \times 10^{-16}$). We also tested TIS reporter in the Abelson-transformed PD-31 cells in two growth conditions: low serum (1%) and in the presence of Imatinib (3uM). The relative translation efficiency was not altered ($R^2 = 0.997$, $p\text{-value} = 2.3 \times 10^{-71}$). These findings are discussed in the text and included in a new figure (Results: paragraph 10, Supplementary Figure 2).

6) Page 7: "Next, we analyzed the initiation efficiency for all human protein-coding TISs... In summary, we found that protein-coding ORFs were enriched for more efficient TIS sequences, while uORFs were not." All the results mentioned in this section should be followed by a p-value; currently, it is hard to interpret them. In addition, the results are clearly not surprising (we expect to see that protein-coding ORFs were enriched for more efficient TIS sequences). It may be interesting (and relatively easy) to check if there is correlation between the translation initiation efficiency of human ORFs (based on the model inferred by the experiment) and their protein levels (or any other measure related to their expression levels).

We agree that the results of this section are in line with previous qualitative findings (especially that protein-coding ORFs are enriched for more efficient TIS sequences). However, for the first time we can move from qualitative descriptions to actual quantitative measurements. Furthermore, we were surprised that uORFs were not enriched for inefficient TIS sequences. To improve the statistical rigor of this section, we have added 95% C.I. to all of our values.

At the reviewer's suggestion, we examined the relationship between TIS efficiency of human ORFs and protein abundance (see figure below). The results showed a very weak relationship between TIS efficiency and protein abundance. TIS efficiency is just one step in the cascade of processes that determine protein abundance. Even in the step of translation, mRNA half-life, ribosome loading rate, elongation rate, and the TIS efficiency all affect the overall translation rate (Maier et al, 2009). We do not feel that these results enhance the manuscript and have therefore omitted them.

TIS efficiency of high and low abundance proteins Proteins were either classified as low abundance (500-698 copies per cell, bottom 5%) or high abundance ($> 5.4 \times 10^5$ copies per cell, top 5%) based on protein copy number in human bone osteosarcoma cell line (U2OS) (Beck et al, 2011). **(A)** Distribution of TIS efficiency values for low and high abundance proteins. High abundance proteins have protein-coding TISs that are $4 \pm 3\%$ more efficient than low abundance proteins. **(B)** Cumulative distribution of TIS efficiency for low and high abundance proteins ($p = 0.14$, Komogorov-Smirnov two-sided test).

7) The subsection "Somatic TIS mutations associated with tumorigenesis": I understand that the mutations introduced in the GFP protein? and are not based on the cancerous gene (or part of it) fused to the GFP? (the second possibility is of course "stronger") In any case, this point should better emphasized. It may be interesting/helpful to compute a p-value related to the effect of the mutations associated with

tumorigenesis on translation initiation (based on their model) vs. the effect of the mutations in this region not associated with tumorigenesis on translation initiation.

We have attempted to make this section more statistically rigorous by (1) adding 95% confidence intervals to our predictions and (2) performing the flow cytometry experiments in triplicate (and adding 95% C.I.).

We have emphasized that we only modified the 11-base TIS sequence (Results: paragraph 16, Methods). Unfortunately it was not clear what the reviewer is requesting regarding the “p-value related to the effect of the mutations associated with tumorigenesis on translation initiation... vs. the effect of the mutations in this region not associated with tumorigenesis.” Any of the mutations may be associated with tumorigenesis. After all, the mutations were originally isolated in tumor samples. However, for a subset of sequences we can state that the effect of the mutation is consistent with known tumor expression patterns. The set of mutations that are not consistent with known tumor expression patterns may still help to drive tumor formation in some unknown mechanism.

8) Methods; subsection "Ribosome footprint profiling analysis": "Scoring of ribosome footprint profiles was performed using a custom MATLAB graphical user interface. The user was displayed a trace of the ribosome footprint data with the location of the annotated TIS and the putative truncation TIS. The user was asked to manually score if the ribosome footprint data supported the putative truncation TIS. As with all visual scoring methods, human bias was a major concern. In order to limit any bias, the transcripts were scored in a random order and with no additional information provided to the user. An attempt was made to automate the TIS peak calling. However, the results were inconsistent with visual inspection and were, in our opinion, unsatisfactory. Because we had a limited number of transcripts to score, we chose a manual process that emphasized accuracy over throughput. All transcript traces, predictions, and scoring are provided in Supplementary Figure 4." This part of the analysis should be changed and based on a computational/mathematical scoring (which should be clearly defined/described) followed by p-values; it is unacceptable to provide results based on manual human dependent analysis that cannot be reproduced in future studies.

We certainly want our results to be reproducible in future studies. We have now provided the entire set of ribosome footprint profiling traces used in this study along with a description of how we scored the traces (Supplementary Figure 7). Any researcher could re-score the traces and compare the results.

Creating an automated scoring algorithm would not remove the human dependent analysis. An automated scoring algorithm would need to be trained against known alternative initiation sites. The list of experimentally confirmed alternative initiation is far too small to properly train the algorithm. Therefore, to create a training set we would need to manually score ribosome footprint profiling traces. Since the algorithm would be based on our manual scoring, it would unfortunately suffer from the same pitfalls as our human dependent analysis and the algorithm would not be as transparent. Furthermore, to generate a proper training set, we would need to manually score a few hundred ribosome footprint traces, however there are only a total of 980 transcripts that are candidates for alternative initiation. Therefore the automated scoring algorithm would be trained on the same data that it was meant to analyze.

9) The section "Quantitative leaky scanning model of alternative initiation". The comparison of the prediction of the translation initiation model to ribosomal profiling data is important; however, this section should be significantly improved. First, I suggest to also check the data of Lee at al. (see point1)); second, all the genes should be analysed (the authors write "we chose to analyze 235 transcripts with predicted initiation ratio, X {greater than or equal to} 0.33 and 235 transcripts whose predicted initiation ratio, X , was < 0.33"); third, also here the authors should provide quantitative measures based on all the ATG codons such as correlations, p-values and error-rates connecting their predictions to the ribosomal density measurements.

In addition, it is not clear to me why the authors did not analyze out of frame downstream AUG start codons and upstream codons, and compared their prediction to the measurements of the ribosomal profiling data.

We initially considered using data from both Lee et al and Ingolia et al. However we decided to use Ingolia data for the primary reason that there were far more ribosome footprint reads. The Ingolia et al dataset had 21,006,597 ribosome footprint reads in the presence of harringtonine. The Lee et al. dataset had only 3,734,064 ribosome footprint reads in the presence of lactimidomycin. Additionally, the effort to process the Ingolia et al data set was considerable. We did not see adequate justification for repeating the process with the Lee et al dataset.

We have analyzed all of the 980 transcripts that meet our filtering criteria. Our original motivation for not including all of the transcripts was that we wanted to focus on transcripts that were likely to have alternative initiation sites ($X > 0.33$), but we also wanted to analyze enough transcripts that were unlikely to contain an alternative initiation site ($X < 0.33$) to properly estimate the false negative rate. Therefore, we only analyzed a fraction of the transcripts where $X < 0.33$. We realize that this is confusing and have now, for the sake of simplicity, analyzed all 980 transcripts. The results have not changed. We have added the p-value when comparing results from our quantitative leaky scanning model to the qualitative -3/+4 rules. Figure 8A-B provides the error-rates associated with both methods (Results: paragraph 21, Figure 8).

The goal of this section was to apply our leaky scanning model to identify N-terminal truncation protein isoforms, which have interesting biological implications and in our opinion have been under studied. Additionally, we could largely ignore the possibility of re-initiation when analyzing TISs downstream and in-frame of the annotated TIS.

Minor

1) Please add to the abstract quantitative measures such as correlations, error rates, relative translation, etc to support the reported conclusions.

The abstract is limited to 175 words and we felt that we could not add these measures at the cost of important statements in our abstract.

2) Subsection "Position Weight Matrices" of the Methods section: I understood the formulas but they can be improved and be written in a more clear/accurate manner. For example, the score of a sequence with nt $b = -6 : +5$ is $k \cdot \exp(\sum_b C_{\{b,i\}})$ where $C_{\{b,i\}}$ is the coefficient of nt i ($i = \{A, T, C \text{ or } G\}$)

We strive to be as clear and accurate as possible. We have followed the reviewers recommendation and explicitly defined $b = \{U, C, A, \text{ or } G\}$ and $i = \{-6, -5, -4, -3, -2, -1, +4, \text{ or } +5\}$. We have also explicitly defined the sum to be over $i = -6 : +5$. (Methods: PWM section)

Response to reviewer #3:

Specific comments:

1. The authors need to introduce the eukaryotic translation process with more mechanistic detail, and provide several examples from the literature to show how changes to the 5' UTR sequence, besides the Kozak sequence, can greatly alter translation rate. The manuscript should clearly indicate that the number of ribosomes that bind to a eukaryotic mRNA is not determined by the Kozak sequence, but is instead controlled by mRNA sequences and structures farther upstream as well the cap-bound protein complex that mediates 43S binding. The Kozak sequence determines the fraction of scanning ribosomes

that select a start codon for translation initiation. This will explain why two mRNAs can have the same Kozak sequence, but in fact can express very different amounts of protein.

The terminology in the article "The initiation of mammalian protein synthesis and mRNA scanning mechanism" by Lomakin & Steitz should be used to describe 43S ternary complex formation, 43S scanning, and formation of the 48S PIC when pausing in front of the Kozak occurs. <http://www.nature.com/nature/journal/vaop/ncurrent/full/nature12355.html>

We have added more mechanistic detail of translation initiation throughout the text.

The first paragraph of the Introduction now reads, "...In eukaryotes, translation initiation typically follows the scanning ribosome model. In this model, the ribosomal preinitiation complex consisting of the small ribosomal subunit, Met-tRNA, eIF2-GTP, eIF1, and eIF1A are loaded onto the mRNA 5' cap (Kozak, 2002b; Jackson et al, 2010; Hinnebusch, 2011). The preinitiation complex, along with additional initiation factors, scans from the mRNA 5' cap in the 3' direction in search of a start codon, which in most circumstances is AUG. When the preinitiation complex recognizes a start codon, initiation factors dissociate and phosphate is irreversibly released. The large ribosomal subunit is then able to join the small ribosomal subunit and translation elongation commences (Pestova & Kolupaeva, 2002; Nanda et al, 2013)..." (Introduction, paragraph 1).

We have tried to emphasize the fact that we are only measuring the probability of a scanning ribosome initiating at a specific TIS, and, as the reviewer pointed out, we are not accounting for the various other factors that affect the translation rate. To highlight this point, we have added the following text "Since the TIS reporters only differed at the TIS sequence, we used reporter output as a measure of the relative translation initiation efficiency of each TIS sequence; however, when interpreting our results, one should still consider that changes to the eight TIS bases could have indirectly affected the many other factors that influence translation, e.g., loading of the ribosome preinitiation complex or the rate of translation elongation." (Results, paragraph 3)

2. Related to #1, the authors must describe the sequence and structure of their 5' UTR because it is essential towards comprehensive analysis and use of their model. This description must include calculations to indicate the presence of stable RNA structures in the 5' UTR, their location, and proximity to 5' methylG cap and to the Kozak sequence.

As requested we have described the 5'UTR in the manuscript. (Results: paragraph 11) "The 5'UTR used in this study was predicted to have a weak hairpin at 5'cap (bases 1-28, $\Delta G = -8$ kcal/mol) and a stable structure in the retroviral U5 region (bases 83-191, $\Delta G = -51$ kcal/mol)."

3. The measurement precision of the FACS-Seq approach is not adequately addressed. First, the distribution of TIS sequence reads in each of the 20 sorted bins is non-Gaussian. Some distributions have very long tails, indicating that the same TIS sequence can cause large changes in translation efficiency, due to some confounding or time-dependent variable. The authors do not provide an explanation, and overly simplify the analysis by only using the median of the distribution to quantify the overall translation efficiency for each TIS sequence. The authors should calculate the variance of the distributions and show via a two-tailed significance test that TIS sequences with a median translation rate of X have statistically different read sequence distributions, compared to TIS sequences with a median translation rate of X+dX, where dX is the maximum precision that could be claimed for the FACS-Seq approach. The authors should then take care to avoid making comparisons where the difference is less than dX as those will not be statistically significant comparisons. When parameterizing models with FACS-Seq data, or comparing model predictions to FACS-Seq measurements, one should not expect a greater accuracy than the FACS-Seq precision unless by fortuitous chance. The authors supplement their FACS-Seq measurements using (more precise) flow cytometry measurements of individual TIS sequences; the same statistical significance testing must be performed for these measurements.

Each of our measurements (i.e., median GFP/RFP ratio) is a combination of the true TIS efficiency plus some error (i.e., noise). For some of the TIS sequences, we obtain hundreds of sequencing reads and therefore can get a very accurate estimate of the TIS efficiency. For other TIS sequences we get only a handful of reads or even 0 reads. For this reason, making a broad statement about how accuracy of the FACS-seq measurements is challenging. Performing a two-tailed t-test on all possible pairs of TIS sequences is also impractical. To address the reviewer's concern, in the manuscript we have discussed this point:

"Ideally, the sequence coverage generated by FACS-seq would be sufficient to generate histograms of high resolution for all TIS sequences. In practice, TIS sequences with repeat regions were sometimes absent or underrepresented (Figure 2B, white values). Because of data noise, we also could not be certain that any single value could precisely represent an initiation efficiency. Analogous to data obtained from microarray analysis of mRNA, the raw data revealed meaningful trends but individual data points may or may not stand on their own (Kerr et al, 2000; Sultan et al, 2002)." (Results: paragraph 4)

Luckily, the sheer amount of data points helps to reduce the noise by fitting a model to the data. Therefore, we can expect that the dinucleotide PWM will be more accurate than the underlying FACS-seq data. This is analogous to using linear regression to estimate a relationship between 2 variables. Assuming that the relationship is roughly linear, the linear regression on a large set of data points will be a better estimator of the true relationship than any of the individual data points. In other words, "By fitting the raw data to a model, our goal was to more accurately estimate the TIS efficiency for every TIS sequence." (Results: paragraph 4). Of course the dinucleotide PWM reduces the effect of noise, but does not eliminate it. This is why we have included 95% confidence intervals for the efficiency predictions for all possible TIS sequences (Supplementary table II).

4. The maximum range of translation rate efficiencies when mutating the Kozak sequence is never reported, and must be reported. Instead, the translation rate efficiencies are quantified in terms of %, which can be misleading. Based on Figure 4, it appears that the maximum change in translation rate efficiency by a mutation in the Kozak sequence will be about 7-fold. However, the reader should not have to guess-imate when reading the manuscript.

The maximum range of TIS efficiencies was 12-fold. We have added to the text: "The range of TIS efficiencies varied 12-fold with the distribution skewed towards efficient initiation, indicating that the majority of possible TIS sequences that contain an AUG start codon result in efficient initiation." (Results: paragraph 12)

5. Two-tailed significance tests should be performed on the comparisons between the PWM models and flow cytometry measurements, shown in Figure 4. By themselves, R² values can be misleading. In this case, the conclusion should not be different.

The R² values are an effective way to compare various statistical models. However a common mistake is to calculate the R² based on the same data used to train the model. We have not done this. The R² values are based on the model agreement with test data that was not used to train the PWMs. In addition to the R² values, we have included the relevant p-values. (Results: Paragraph 5/6)

6. A key result from the FACS-Seq measurements is that the consensus Kozak sequence is RYMRMVAUGGC. However, a gaping unanswered question is Why? Is there a mechanistic reason why this particular Kozak sequence, which is longer and more specific than previously proposed ones, maximizes the translation rate efficiency? Keeping in mind that the FACS-Seq measurements were performed in only one cell line, under one growth condition, using an existing upstream 5' UTR sequence that was not perturbed, and using one set of fluorescent protein reporters, the question of why becomes more important as there could be non-observed, unaccounted changes in a confounding variable that

becomes the true reason for the extended consensus sequence. If there was corroborating evidence from the literature that this particular Kozak sequence has stronger interactions with the 43S ribosome, then it would answer these questions suitably. Otherwise, the additional "important" nucleotides might only be important in this specific 5' UTR, and not others in the human genome.

Figures 2 and 3 in the article by Lomakin & Steitz show the 43S (scanning) and 48S (paused) ribosomes in complex with mRNA, including the Kozak: Figure 2b shows a crystal structure of the 43S ribosomal complex with initiation factors, initiator tRNA, and mRNA bound together. The mRNA's start codon and surrounding sequences interact with specific ribosomal RNA helices. These interactions cause rotation of the ribosome's head domain (pausing), which then initiates 60S recruitment and translation elongation. Figure 3 shows a crystal structure of the 48S ribosomal complex in its paused state (after rotation) and the contacts with the Kozak sequence that were necessary to achieve head rotation. The section in Lomakin & Steitz's analysis, entitled "The mRNA channel and the mRNA path" discusses the role of the Kozak sequence in pausing. The authors should use the crystal structure to explain why nucleotide mutations from -6 to +7 alter the probability of pausing. They have the opportunity to use some molecular modeling to explain how these nucleotide mutations alter the interactions inside the binding pocket.

Ideally, we would have been able to provide a better biophysical explanation for our observed trends in TIS efficiency. We have even been in contact with Ivan Lomakin regarding his paper on the crystal structure of the 48S PIC. Unfortunately, he confirmed that the resolution of crystal structure is too low to determine the orientation of the residues or bases. Without these orientation, we can only say specific parts of the 48S are proximal to certain TIS positions, not the nature of these interactions. We added the following text to the manuscript: (Discussion, paragraph: 4)

"The biophysical details of how the TIS sequence affects the ribosomal preinitiation complex remain unclear. Recently, Lomakin and Steitz (2013) determined the crystal structure of the mammalian preinitiation complex (48S PIC). The structure confirmed that the -6 to the +7 positions were within the ribosome's mRNA channel, allowing for the possibility that the +6 and +7 positions could also influence initiation. Downstream of the start codon, the +4 and +5 positions were proximal to 18S helix h44, eIF1, and the basic loop of eIF1A. Upstream of the start codon, the ribosomal proteins rpS26e and rpS28e were proximal to the influential -4, -3, and -2 positions. Additionally, the -3 position has been shown to UV-cross-link with eIF2 α , which was not shown in the crystal structure (Pisarev et al, 2006). Interestingly, the suboptimal -3U was cross-linked less efficiently with eIF2 α than a -3G. It is possible -2 and -4 positions cooperate with the influence initiation by promoting the -3 interaction with eIF2 α . Unfortunately, the preinitiation complex crystal structure resolution was too low to determine the orientation of the mRNA bases or the orientation of the residue side chains. A higher resolution crystal structure would enable us to determine not just which parts of the preinitiation complex are proximal to the TIS, but also the nature of these interactions, enabling a biophysical explanation for the TIS efficiency trends described here."

7. The authors neglect the role of mRNA structures that could affect start codon selection and translation efficiency. First, mutation of Kozak sequences could affect the formation of mRNA structures in distant locations, farther upstream of the Kozak. Upstream mRNA structures have been implicated in preventing 43S ribosome from binding the methylG cap, and pausing the 43S ribosome during scanning (PMC1440912). It is also possible that pausing of the 43S by a mRNA structure at the appropriate position over a start codon could actually increase start codon selection. The authors should analyze the compendium of mRNA structures that form in their 65,636 sequences and determine whether low or high translation rate efficiency was instead controlled by the presence or absence of mRNA structures upstream or downstream of the start codon.

The reviewer makes an excellent point. We did analyze the compendium of mRNA structures, however we did not find any relationship between mRNA folding energy and TIS efficiency. To reflect these findings, we have added Supplementary Figure 3 and the following text

"It was also possible that specific TIS sequences could have resulted in the formation of an mRNA structure near the start codon. To investigate this possibility, we calculated the folding energy for 65,536

mRNA sequence associated with each TIS sequence in our reporter library. Each sequence consisted of the 70 bases upstream of the TIS, a single 11 base TIS region, and 70 bases downstream of the TIS (Supplementary Figure 3). The difference between the most stable and least stable secondary structure was -16 kcal/mol. We did not observe any significant relationship between the mRNA folding energy and the TIS efficiency ($p = 0.18$).” (Results: paragraph 11)

8. The statistics in the section on genome-wide TIS predictions are not accompanied by any significance testing. If I flip 10 coins and receive 15% more heads than the average, that can happen by pure chance. If I flip 10000 coins, and receive the same result, that would be significant. Here, the results are closer to the latter, but the two-tailed p-value needs to be calculated. Because the distribution of translation rate efficiencies is likely non-Gaussian, it also helps to include a Kullback-Leibler Divergence plot in the Supp. Info to compare random to actual translation rate distributions.

We have improved the statistical rigor of this section. We have included a Kullback-Leibler plot as requested (Supplementary Figure 5) and have also included 95% confidence intervals to all of our values. The difference in mean efficiency between the uORF TISs and the TIS sequence space is only $1.6 \pm 0.4\%$ (95% C.I.). This value is statistically significant ($p = <2e-16$) but we is of arguable practical consequence.

9. In referring to Grave's disease, the sentence "The polymorphism was found to decrease the TIS efficiency by 16%, demonstrating that even modest changes ... can be relevant" cites only the in vitro result from Jacobson et. al. When they repeated CD40 expression measurements in vivo, they observed 32% less surface CD40 expression in fibroblasts, and 39% less surface CD40 surface expression in B cells. The difference between 16% and 32% is significant when the authors' follow-up analysis reports differences of 27% to 46% as biologically meaningful results, which may be false if one objectively considers the existing literature.

We have changed to the text to read “The polymorphism was found to decrease the TIS efficiency by 15-32%, ...” This is the range includes both the in vitro and the in vivo results and is the range that the authors chose to cite in a follow up paper when referring to the effect of the C/T mutation at the -1 position (Jacobson et al, 2007, Genes and Immunity). We still feel that these values support the statement that “modest changes in TIS efficiency can be biologically relevant.” Or as Jacobson et al (2005) stated themselves, “In light of our findings, we suggest that pathogenesis of [Graves' Disease] might be influenced by translational pathophysiology, where even modest change in the efficiency of translation of an mRNA transcript may manifest itself in the development of, or predisposition to, disease.”

10. Figure 7 does not have error bars and no statistical significance testing has been performed. This is particularly troubling since the changes in expression with vs. without the mutation are less than 50% for half of the TIS sequences, and the difference between model predictions and measurements appears to be greater than 20% for two of the TIS sequences. The usage of percentage as a unit further obscures the differences between the actual fluorescence data. The authors must perform these measurements in triplicate and carry out two-tailed significance tests.

We have performed the experiments in triplicate and added error bars to both the predicted and observed values (Figure 7). It is important to remember that prior to this manuscript, qualitative predictions about the effect of these mutations would have been difficult. We are able to make quantitative predictions. While there may be some error in our quantitative predictions, it is a vast improvement over previous qualitative predictions.

11. There is a certain amount of tautology in the analysis of the small number (7) of somatic cell mutations that were analyzed. The authors "chose mutations where the predicted change in protein translation efficiency was consistent with known tumor expression patterns", and then concluded that "In summary, we have linked changes in TIS efficiency with known tumorigenic expression patterns". The authors can address this concern by expanding their analysis not only to SNPs found in TIS sequences

related to tumorigenesis, but to TIS sequence SNPs that are associated with any disease condition. The authors should strive to present a sufficiently large number of validated predictions to convince the reviewers that the model does not have confounding variables. While a p-value < 0.05 is the standard bare minimum, it is not persuasive.

In our opinion, our findings do not suffer from a tautological error. The known tumorigenic expression patterns were based on changes in gene copy number and mRNA transcript level; the expression patterns were completely independent of the translation efficiency. Prior to this work there was no evidence that these mutations increased or decreased the TIS efficiency. To emphasize this point, the text now reads:

"In summary, we have linked known tumorigenic expression patterns, previously validated only for RNA level or gene copy number, with changes in TIS efficiency, thereby providing a candidate explanation for how TIS mutations could promote tumorigenesis." (Results: paragraph 17)

Furthermore, our claims were not tautological because they could have been falsified. If we predicted a mutation to affect TIS efficiency in a manner consistent with known tumorigenic expression patterns, but our experiments found the TIS efficiency to change in the opposite direction, then our predictive model would not be of much use. Lastly, we want to point out that we chose to use the word "link" in our statement. This wording indicates an association but does not conclude a causative relationship, which can only be presumed at this point.

12. Regarding the leaky scanning model section, there is no explanation why 980 transcripts fit the authors' criteria, but only 470 transcripts were analyzed. The authors should analyze the 980 transcript data-set.

We absolutely agree and have analyzed all 980 transcripts. The results have not changed. See response to comment #9 from reviewer #1.

13. Overall, the section on the leaky scanning model repeats existing ribosome profiling data; using the PWM model does not explain any surprising observations or contribute to further understanding. It would be more interesting if low-efficiency nucleotides in the extended Kozak sequence, and not just the original Kozak sequence, were responsible for a truncation TIS that was biologically relevant.

Determining which truncations are biologically relevant is beyond our immediate capabilities. (Although one could argue that the very fact that a truncation TIS occurs means that it is biologically relevant.) However, the fact that the quantitative leaky scanning model, which incorporates information about the entire TIS sequence, performs better than the qualitative -3/+4 rules demonstrates that bases other just the -3 and +4 positions are responsible for truncation TISs.

14. The sequence space of biological motifs grows combinatorially, not exponentially, larger with each additional base. Exponentially is e^n . In this case, combinatorial is 4^n . $4 > e$.

Exponential growth is technically any situation where the rate of growth is proportional to the current size, i.e. $df/dx = c*f(x)$. This means that any equation with the form $f(x) = k^{cx}$ represents exponential equation, regardless of the value of k. In fact, we could change the exponential base. For example, $f(n) = 4^n$ is the same as $f(n) = e^{\ln 4^n}$, where n is the number of bases and f(n) is the number sequences. Therefore, even under the reviewer's definition of exponential growth, we believe we would still be correct in stating that the TIS sequence space grows exponentially larger with each additional base.

15. The sentence "Our ability to analyze such a large number of sequences with relatively high resolution" is not supported by the absence of measuring the precision of the FACS-Seq approach or the quantitative reporting of the translation rate range that was observed.

We altered the sentence to read, "Our ability to analyze such a large number of sequences was dependent on the precision of the readout from our translation reporter." (Discussion, paragraph 6)

16. The authors propose FACS-Seq as a technique for characterizing large sequence spaces. However, they do not provide any caveats or limitations to the FACS-Seq approach. Certainly, the authors would agree that FACS-Seq can not characterize 10^{12} sequences in a single-run, and yet this is the number of sequences in a 20 nucleotide motif. With these constraints, what are the most intelligent applications of FACS-Seq? Can one "predict the behavior of all sequence elements employed in the genome" using only FACS-Seq and informatic relationships?

While FACS-seq can analyze very large library sizes, even larger than the one analyzed in this manuscript, the reviewer is absolutely correct in stating that FACS-seq has limitations on the library size. We did not mean to mislead the reader and have added to the following text to address this issue: "To our knowledge, our implementation of FACS-seq has to date analyzed the largest number of sequences in one experiment (65,536 vs. 14,234 (Goodman et al, 2013)). With some optimization, we believe that FACS-seq can be applied to libraries approaching 10^6 sequences, with the practical limit being determined by the time required to sort the library." (Discussion, paragraph 6)

17. The cell lines utilized need to be described in the main text. These are not only methodological details, but could affect the central conclusions of the manuscript.

We have described the PD-31 cell line in the main text (Results: paragraphs 2 and 11). Additionally, we tested a subset of our TIS library in 5 additional cell lines to demonstrate that the central conclusions of this manuscript are not specific to the PD-31 cell line (Supplementary Figure 2, Results: paragraph 11).

18. It was also troubling that "linear regression was performed using the natural logarithm of the raw FACS-seq data as the dependent variable". In Salis et. al., the relationship between the ribosome's binding free energy to a mRNA was log-linear because, according to statistical thermodynamics, the probability of ribosome binding was proportional to $\exp(-dG/RT)$, where dG is the amount of reversible work needed to bind. Here, is there any expectation, based on the scanning mechanism, that the efficiency of start codon selection would be related to the logarithm of protein fluorescence? No, there should be a linear relationship between the probability of start codon selection and the resulting protein fluorescence. Because this is a purely informatic relationship, log-transforming protein fluorescence will simply alter the parameterized coefficients. The authors should repeat the parameterization using protein fluorescence as the dependent variable and receive other-valued coefficients. These coefficients would likely give you a similar PWM prediction. This is an example of why the numbers in informatic models have no physical or biological meaning.

The justification for using a log-linear relationship has been worked out in detail in Barrick et al, 1994. Furthermore, the ribosomal preinitiation complex reversibly interacts with the mRNA. For example, Lomakin and Steitz (2013) hypothesized that the reversible interaction between the -3 position and the ribosomal rps26e can promote initiation. Our aim in log transforming the fluorescence data prior to fitting the model was to keep the model as close as possible to the underlying biophysical processes. The reviewer is correct in that a model without any biological basis could have performed approximately equally as well. For example, we achieved similar results using more advanced statistical modeling techniques such as random forests, and LASSO regression. However we chose to use the model that we believe to reflect the underlying biophysical process of initiation (i.e., weak, reversible interactions between the mRNA and ribosomal complex).

Thank you again for submitting your work to Molecular Systems Biology. We have now heard back from the two referees who agreed to evaluate your manuscript. As you will see from the reports below, the main concerns of both reviewers have been satisfactorily addressed. However, reviewer #1 lists a few relatively minor issues that we would ask you to address in a revision of the manuscript.

Reviewer #1:

The authors performed most of my suggestions and significantly improved the paper. I have a few additional minor suggestions for further improvements.

1. All the data generated in this study (e.g. TIS efficiencies of all TIS sequences in all the cases) should be published as a supplementary .xls table (all data needed for reproducing the reported results/graphs/etc should be included). This will enable comparisons to these data in future studies.

2. Point 6. In my previous review: the authors did not find significant relation between TIS efficiency of human ORFs and protein abundance. Though this is a negative result it should be published (you can put the details in the supplementary). The result can teach us about the rate limiting steps in translation and the effect of TIS efficiency on protein levels in human endogenous proteins.

3. Point 7. of reviewer 2: the fact that there is no correlation between mRNA folding induced by the TIS sequence and TIS efficiency. I suspect that there will be a correlation if the folding is computed based on shorter windows (e.g. 15 bases upstream the TIS + TIS 11 base TIS region + 15 bases downstream the TIS) as was done in previous studies (see for example, PMID:20140241, PMID:20133581, PMID:24072823) for previous organisms/genes; could you check this point. In any case, I suggest to mention the previous studies that did find a correlation.

Reviewer #3:

The authors have carefully responded to this reviewer's comments with acceptable additions and changes to their manuscript. The modified manuscript now reads as a more carefully executed set of measurements and analysis that should be re-usable for many applications.

We are submitting a second revision of our paper. We were pleased that the reviewers were satisfied with our previous revisions. The reviewers' remaining issues are now addressed and a point-by-point response can be found below. Again, we thank the reviewers for their thoughtful comments.

Lastly, with your permission, we would like to change our title to "Quantitative analysis of mammalian translation initiation sites by FACS-seq", where we have added "by FACS-seq" to better highlight the novel method we used in our high-throughput analysis.

Response to reviewer #1:

The authors performed most of my suggestions and significantly improved the paper. I have a few additional minor suggestions for further improvements.

1. All the data generated in this study (e.g. TIS efficiencies of all TIS sequences in all the cases) should be published as a supplementary .xls table (all data needed for reproducing the reported results/graphs/etc should be included). This will enable comparisons to these data in future studies.

Following the MSB guidelines we have included “Source Data” allowing for the reproduction of our figures.

2. Point 6. In my previous review: the authors did not find significant relation between TIS efficiency of human ORFs and protein abundance. Though this is a negative results it should be published (you can put the details in the supplementary). The result can teach us about the rate limiting steps in translation and the effect of TIS efficiency on protein levels in human endogenous proteins.

We believe that these results are beyond the scope of this paper and have, therefore, omitted them from the manuscript. The original reviewer comment requested that we “check if there is correlation between the translation initiation efficiency of human ORFS ... and their protein levels.” We made a good faith effort to check for such a correlation, but found only a very weak correlation. As translation initiation is only one of the biological processes that determines protein concentration, it is not surprising to us that we did not observe a strong correlation. It is possible that a more rigorous model of protein abundance that incorporates translation initiation efficiency as well as mRNA translation rates, mRNA half-lives, upstream ORFs, and protein half-lives would yield more significant results. While such a model is beyond the scope of this paper, we may follow up on these results in a future publication.

3. Point 7. of reviewer 2: the fact that there is no correlation between mRNA folding induced by the TIS sequence and TIS efficiency. I suspect that there will be a correlation if the folding is computed based on shorter windows (e.g. 15 bases upstream the TIS + TIS 11 base TIS region + 15 bases downstream the TIS) as was done in previous studies (see for example, PMID:20140241, PMID:20133581, PMID:24072823) for previous organisms/genes; could you check this point. In any case, I suggest to mention the previous studies that did find a correlation.

First, we followed the reviewer’s suggestion to determine if our results were an artifact of our window size. We repeated our analysis with a shorter window consisting of 15 bases upstream, an 11 base TIS region, and 15 bases downstream. We found that the window length does not have any significant effect on our results. The results with the original 151 base window were consistent those from the shorter 41 base window (151 base window: $R^2 = 0.02$, $p = 0.18$; 41 base window: $R^2 = 0.02$, $p = 0.80$).

Second, two of the three studies that the reviewer cited (PMID:20133581, PMID:24072823) dealt only with *E. coli* and/or *S. cerevisiae*. In the paper, we discuss the possibility that prokaryotes and simple eukaryotes are not capable of efficiently resolving secondary structures near the start codon, while the mammalian mRNA helicase, eIF5, prevents secondary mRNA structures from interfering with translation initiation. Comparing the yeast and mammalian TIS consensus sequence, one possible explanation for the divergence of the sequences is that the high-AU content of yeast mRNA prevents secondary structures (Discussion, paragraph 2). The third study cited by the reviewer (PMID:20140241) studied a variety of species and, when considering prokaryotes and simple eukaryotes, obtained results consistent with the 2 studies cited above. However, for warm-blooded species, the authors did not observe a preference for reduced mRNA secondary structure stability near the start codon (Figure S1). Additionally, for humans the authors did not observe a relationship between mRNA secondary structure and gene expression (Figure 6). These results agree with our findings that mRNA secondary structure has minimal, if any, impact on the efficiency of initiation in mammals. We have referenced all three of the previous studies in our manuscript.

Reviewer #3:

The authors have carefully responded to this reviewer's comments with acceptable additions and changes to their manuscript. The modified manuscript now reads as a more carefully executed set of measurements and analysis that should be re-usable for many applications.